# PERCEPTION-AWARE POLICY OPTIMIZATION FOR MULTIMODAL REASONING

**Zhenhailong Wang[1][*][†], Xuehang Guo[1][*], Sofia Stoica[1], Haiyang Xu[2][†], Hongru Wang[1],**
**Hyeonjeong Ha[1], Xiusi Chen[1], Yangyi Chen[1], Ming Yan[2], Fei Huang[2], Heng Ji[1][†]**
[1]University of Illinois Urbana-Champaign    [2]Alibaba Group
[*]Equal contribution    [†]Corresponding author
`wangz3@illinois.edu, shuofeng.xhy@alibabainc.com, hengji@illinois.edu`

## ABSTRACT

Reinforcement Learning with Verifiable Rewards (RLVR) has proven to be a highly effective strategy for empowering Large Language Models (LLMs) with long chain-of-thought reasoning abilities. However, its design and optimizations remain tailored to purely textual domains, resulting in suboptimal performance when applied to multimodal reasoning tasks. In particular, we observe that a major source of error (67%) in current multimodal reasoning lies in the perception of visual inputs. To address this bottleneck, we propose Perception-Aware Policy Optimized (PAPO), a novel policy gradient algorithm that encourages the model to learn to perceive while learning to reason. Specifically, we introduce the Implicit Perception Loss in the form of a KL divergence term, which can be seamlessly plugged into mainstream RLVR algorithms such as GRPO and DAPO. Notably, PAPO does not rely on any additional data annotation, reward models, or stronger teacher models. Despite its simplicity, PAPO yields significant overall improvements of 4.4%-17.5% on eight multimodal reasoning benchmarks. The improvements are more pronounced, approaching 8.0%-19.1%, on tasks with high vision dependency. We also observe a substantial reduction of 30.5% in perception errors, indicating improved perceptual capabilities with PAPO. Overall, PAPO offers a new perspective on advancing multimodal RLVR via the optimization objective, moving beyond rollout or reward design and pointing toward deeper integration of perception and reasoning.

## 1 INTRODUCTION

Reinforcement Learning with Verifiable Rewards (RLVR) has emerged as a key driver of recent progress in large language models (LLMs), particularly in enhancing their reasoning capabilities. By optimizing models using verifiable signals, such as structured thinking formats and final answer accuracy, RLVR has demonstrated strong empirical success in models like DeepSeek-R1 (Guo et al., 2025), as well as through algorithmic innovations like Group Relative Policy Optimization (GRPO) (Shao et al., 2024). Large Multimodal Models (LMMs), however, continue to struggle with complex multimodal reasoning tasks that require both fine-grained perception and multi-step reasoning (Yue et al., 2024; Zhang et al., 2024; Wang et al., 2025e). This limitation stands in contrast to the strong reasoning performance of LLMs in textual domains.

Aiming to address this gap, a growing body of work (Chen et al., 2025; Shen et al., 2025a; Meng et al., 2025b; Huang et al., 2025; Yang et al., 2025; Liu et al., 2025a; Wang et al., 2025d; Xiao et al., 2025a; Zhu et al., 2025b; Wang et al., 2025b; Liang et al., 2025; Xia et al., 2025; Xiao et al., 2025b; Shen et al., 2025c; Wan et al., 2025; Yao et al., 2025) has explored applying RLVR to LMMs in hopes of similarly improving their multimodal reasoning abilities. Initial successes have been reported, particularly in terms of generalization ability when using GRPO compared to supervised finetuning (Chen et al., 2025; Shen et al., 2025a; Huang et al., 2025). However, most prior efforts have primarily focused on improving **data** and **rollout** quality (Li et al., 2025a; Liang et al., 2025; Wang et al., 2025b; Li et al., 2025b; Liu et al., 2025a; Yao et al., 2025) or **reward** design (Xiao et al., 2025a; Xia et al., 2025; Li et al., 2025c) leaving the core optimization objective largely unchanged

from its application in textual domains. This raises two fundamental research questions: **(1)** *Are there unique challenges in multimodal reasoning that do not arise in text-only settings and cannot be addressed solely through data- or reward-level modifications?* **(2)** *If so, how can we address this by designing a new RLVR **optimization objective** that is better grounded in multimodal domains?*

To investigate the first question, we conducted a comprehensive error analysis on a multimodal reasoning model trained using the standard GRPO pipeline. We manually examined 200 error cases across four benchmarks and categorized the types of errors. Surprisingly, as shown in Figure 1, we found that 67% of the errors stemmed from perception (see § 2.2 for more details). We attribute this bottleneck to the fact that existing RLVR objectives do not explicitly incentivize models to generate visually grounded responses. Recent approaches (Xia et al., 2025; Xiao et al., 2025a; Li et al., 2025c) have also recognized the importance of perception, introducing additional rewards that either directly assess perception quality or require the model to explicitly perform captioning before reasoning. While promising, these strategies often impose a rigid separation between perception and reasoning, rather than enabling joint learning of both. They also rely on additional large neural-based reward models, resulting in significant computational overhead and limitations imposed by the reward model's capacity.

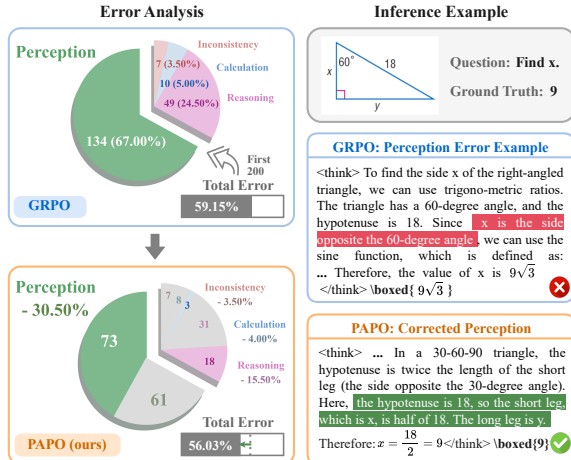

Figure 1: Comprehensive error-type breakdown and inference example between GRPO and PAPO. We observe that perception errors account for the majority (67%) of failures in current multimodal reasoning models trained with GRPO. PAPO significantly reduces the dominant perception-driven errors by 30.5%, with the reduced portion indicated in gray. On the right, we present an inference example illustrating how enhanced perception enables better reasoning.

In this work, we challenge the prevailing view that multimodal reasoning in RLVR can be addressed solely through data, rollout, or reward modifications. Instead, we investigate a deeper and more efficient integration of perceptual incentives into the core optimization objectives. To this end, we propose **Perception-Aware Policy Optimized (PAPO)**, a novel policy gradient algorithm that enhances multimodal reasoning through visually grounded optimization. Notably, PAPO can serve as a direct drop-in replacement for GRPO (Shao et al., 2024) or DAPO (Yu et al., 2025).

The key idea behind PAPO is to encourage the model to learn to perceive while learning to reason. Intuitively, a well-learned multimomdal model should rely on informative visual context while performing reasoning. Based on this intuition, we introduce a Kullback-Leibler divergence (KL) objective, **Implicit Perception Loss (KL$_{\text{prcp}}$)**, which we *maximize* within an RLVR framework. As illustrated in Figure 2, this "reverse KL" loss is computed between two probability distributions over the **same** rollout token sequence, conditioned on either the original or the corrupted visual inputs. From an information gain (Shannon, 1948) perspective, this loss encourages the model to assign higher probability to the response when more informative visual input is provided. As a result, we are able to incentivize the generation of visually grounded responses without requiring any external supervision. To better regularize the unbounded KL$_{\text{prcp}}$ objective, we further introduce a **Double Entropy Loss**, which effectively enhances training stability without compromising performance.

Despite its simplicity, PAPO delivers consistent improvements over GRPO and DAPO across eight multimodal reasoning benchmarks with an average gain of 4.4%-17.5%. The improvement is particularly pronounced (8.0%-19.1%) in tasks with higher vision-dependency where the input question provides limited visual clues. Furthermore, we observe a significant 30.5% reduction in perception-related errors with PAPO, as evidenced by the manual analysis shown in Figure 1. Finally, PAPO also shows faster convergence with early-stage gains starting around 25 steps.

To summarize, our main contributions are threefold: (1) We present PAPO, a new policy optimization algorithm that encourages the model to generate visually grounded responses. To our knowledge, this is the first work to explore a deeper integration of perception-aware supervision signals beyond reward-

level modifications. (2) Comprehensive evaluations across varying levels of vision dependency show consistent improvements of PAPO over GRPO and DAPO, using identical training data and reward functions. (3) We conduct extensive analyses of PAPO and identify potential training instabilities, which we mitigate through entropy-based losses.

## 2 PRELIMINARY

### 2.1 GROUP RELATIVE POLICY OPTIMIZATION (GRPO)

GRPO (Shao et al., 2024) is a variant of the Proximal Policy Optimization (PPO) (Schulman et al., 2017) algorithm that removes the value model and estimates advantages via group-based computation. In the context of multimodal reasoning, consider a dataset $D$ containing datapoints consisting of visual inputs $I$, questions $q$, and ground truth answers $a$. The GRPO learning objective with respect to the policy $\pi_\theta$ can be written as follows, where $\theta$ represents the parameters in a large multilmodal model:

$$\mathcal{J}_{\text{GRPO}}(\theta) = \mathbb{E}_{[\{o_i\}_{i=1}^G \sim \pi_{\theta_{old}}(O|q,I)]} \frac{1}{G} \sum_{i=1}^G \frac{1}{|o_i|} \sum_{t=1}^{|o_i|} \Big\{$$

$$\min\Big[r_{i,t}(\theta)\hat{A}_{i,t}, \text{clip}\left(r_{i,t}(\theta), 1-\epsilon_l, 1+\epsilon_h\right)\hat{A}_{i,t}\Big] - \beta\mathbb{D}_{KL}\left[\pi_\theta||\pi_{ref}\right]\Big\}$$

$$\text{with } r_{i,t}(\theta) = \frac{\pi_\theta(o_{i,t}|q,I,o_{i,<t})}{\pi_{\theta_{old}}(o_{i,t}|q,I,o_{i,<t})} \tag{1}$$

$G$ denotes the size of the group which contains multiple responses $O$ sampled from the rollout policy $\pi_{\theta_{old}}$, corresponding to one input instance $(q, I)$. $\epsilon_l, \epsilon_h \in R$ are hyperparameters for clipping too large updates. The original GRPO (Shao et al., 2024) sets $\epsilon_l = \epsilon_h$, while recent work (Yu et al., 2025) shows benefits of clip-higher, i.e., $\epsilon_h > \epsilon_l$. We follow the clip-higher configuration in all of our experiments. The token-level advantage $\hat{A}_{i,t}$ is defined as the sequence-level reward $\widetilde{R}_i$ normalized across the group. Given a reward verifier eq(), which checks whether a response is equivalent to the ground truth, the advantage is computed as follows:

$$\hat{A}_{i,t} = \widetilde{R}_i = \frac{R_i - \text{mean}(\mathbf{R})}{\text{std}(\mathbf{R})}, \text{ where } R_i = \begin{cases} 1.0, & \text{if eq}(a, o_i), \\ 0.0, & \text{otherwise.} \end{cases}$$

where $\mathbf{R} = \{R_1, R_2, \dots, R_G\}$ is the rewards for the current group.

Decoupled Clip and Dynamic Sampling Policy Optimization (DAPO) (Yu et al., 2025) is a representative follow-up to GRPO, introducing several modifications such as Clip-Higher, Dynamic Sampling, and Token-Level Policy Gradient Loss. We refer readers to the original paper for detailed descriptions. In this work, we investigate the application of PAPO to both GRPO and DAPO.

### 2.2 ERROR ANALYSIS OF MULTIMODAL REASONING

We first investigate the question: Are there unique challenges in multimodal reasoning that do not arise in text-only settings? We follow a typical GRPO pipeline to train Qwen2.5-VL-3B (Qwen Team, 2024a) on ViRL39K (Wang et al., 2025b) (experimental details can be found in §4) and manually examine and categorize error types based on 200 error instances sampled from four benchmarks: Geometry3K (Lu et al., 2021), MMK12 (Meng et al., 2025b), LogicVista (Xiao et al., 2024), and MathVerse (Zhang et al., 2024). We identify four dominant error types: **(1) Perception Error**: Inaccurate interpretation of the visual content. For example, in Figure 1, the model associates $x$ with the wrong side; **(2) Reasoning Error**: Mistakes in the logical inference process, such as applying incorrect rules or theorems; **(3) Calculation Error**: Mistakes in performing arithmetic operations; **(4) Inconsistency Error**: Discrepancies between intermediate reasoning steps and the final answer.

We show the error distribution in Figure 1. To our surprise, we find that the majority of errors, 67.0%, stem from poor perception. In many cases, the model performed well in logical or algebraic reasoning but failed to accurately interpret visual inputs, such as spatial relationships or label associations. We attribute this bottleneck in perception to the GRPO objective not providing any incentive for the model to generate visually grounded responses. This leads us to a key question: *can we jointly improve perception and reasoning in multimodal RLVR?* We present our approach in the next section.

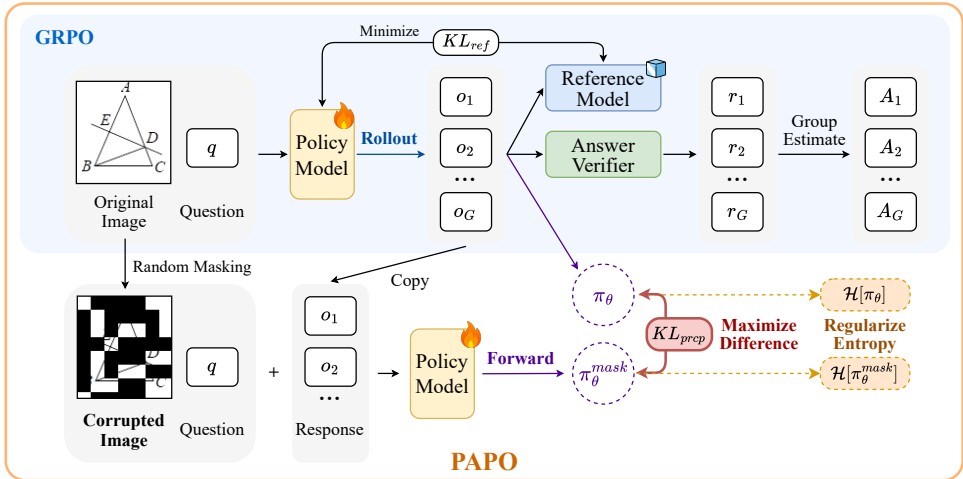

Figure 2: **Illustration of the PAPO$_G$ objective**, which extends GRPO by adding the Implicit Perception Loss (KL$_{\text{prcp}}$). Additional Double Entropy Loss regularization ($H[\pi_\theta]$, $H[\pi_\theta^{mask}]$) can be added for enhancing training stabilities. The KL$_{\text{prcp}}$ is formulated as *maximizing* the difference between the original policy $\pi_\theta$ and a corrupted policy $\pi_\theta^{\text{mask}}$, computed with a masked visual input. Intuitively, PAPO encourages the model to produce visually grounded responses while still achieving high rewards.

## 3 METHOD

### 3.1 TASK FORMULATION

We follow a typical RLVR setting (Shao et al., 2024; Yu et al., 2025), where the training dataset $D$ contains visual inputs $I$, questions $q$, and short ground truth answers $a$. A simple rule-based verifier (hiyouga, 2025) is used to assign rewards for each rollout during training. We do not rely on any existing chain-of-thought data and initiate RL training directly without supervised fine-tuning. **In the following method sections, we use GRPO as an example to elaborate on the PAPO algorithm.**

### 3.2 PAPO

To address the aforementioned unique challenges in multimodal RLVR, we propose **Perception-Aware Policy Optimized (PAPO)**. The key idea behind PAPO is to encourage the policy to **prefer visually grounded responses that can achieve high rewards**. PAPO requires no additional annotations, no reliance on stronger teacher models, and no expensive neural reward models. We formally describe the key components of the PAPO algorithm as follows. Figure 2 shows an illustrative overview of the algorithm.

**Implicit Perception Loss (KL$_{\text{prcp}}$).** To indicate whether a generated response depends on meaningful visual information, we define the following ratio: $r^{\text{prcp}}(\theta) = \frac{\pi_\theta(o|q,I)}{\pi_\theta(o|q,I_{\text{mask}})}$, where $o$ is a generated sequence of tokens, $q$ is the question and $I$ is the original visual input. And $I_{\text{mask}}$ is defined as a *corrupted* visual input, which is constructed by masking out a sufficiently large portion of the original input. Figure 2 shows an example of $I_{\text{mask}}$ where 60% of the patches are masked.

From an information gain (Shannon, 1948) perspective, this ratio quantifies the degree to which the model's output distribution changes when meaningful visual information is removed. A higher ratio indicates that the model assigns significantly lower probability to the correct output when deprived of full visual context, suggesting that the visual input contributes substantial information to the decision. Conversely, a low ratio implies that the model's prediction remains largely unaffected by masking, indicating that it may rely primarily on the textual input rather than truly grounded visual understanding. Thus, intuitively, for a well-behaved multimodal policy model $\theta$, we want $r^{prcp}(\theta)$ to be *high*. Based on this intuition, we introduce an additional loss to the GRPO objective, the Implicit

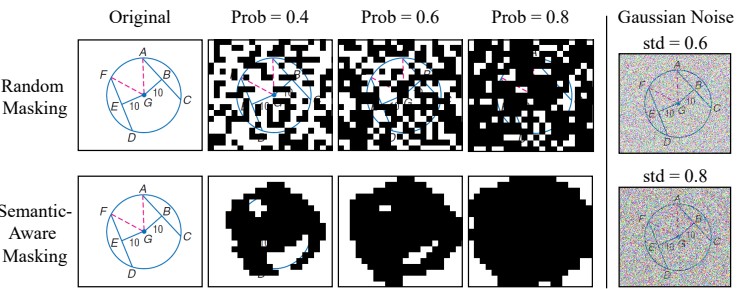

Figure 3: Visualization of different masking strategies. Random masking selects patches uniformly, whereas semantic-aware masking prioritizes patches containing salient objects. Gaussian noise is less effective at obscuring informative semantics, even when applied with a high noise factor.

Perception Loss ($KL_{prcp}$), which is implemented by *maximizing* the following Kullback-Leibler (KL) divergence: $\mathbb{D}_{KL}[\pi_\theta||\pi_\theta^{mask}] = \mathbb{D}_{KL}[\pi_\theta(o|q, I) \parallel \pi_\theta(o|q, I_{mask})]$.

**Entropy Regularization.** Since we maximize a KL divergence that is theoretically unbounded, the model may "hack" $KL_{prcp}$, eventually leading to performance collapse. To further enhance the training stability of PAPO, we introduce **Double Entropy Loss**, an effective regularizer that prevents collapse while preserving performance. This idea stems from our observation that rising rollout entropy in both $\pi_\theta$ and $\pi_\theta^{mask}$ is a representative sign of collapse. Double Entropy Loss encourages the model to keep both entropy values low.

Combining the Implicit Perception Loss and Double Entropy Loss with the GRPO objective yields the complete **PAPO$_G$ objective**:

$$
\mathcal{J}_{PAPO_G}(\theta) = \mathbb{E}_{[\{o_i\}_{i=1}^G \sim \pi_{\theta_{old}}(O|q,I)]} \frac{1}{G} \sum_{i=1}^{G} \frac{1}{|o_i|} \sum_{t=1}^{|o_i|} \Bigg\{
$$
$$
\min \left[ r_{i,t}(\theta)\hat{A}_{i,t}, \text{clip}\left(r_{i,t}(\theta), 1-\epsilon_l, 1+\epsilon_h\right)\hat{A}_{i,t}\right] - \beta\mathbb{D}_{KL}\left[\pi_\theta||\pi_{ref}\right]
$$
$$
+ \gamma\mathbb{D}_{KL}[\pi_\theta||\pi_\theta^{mask}] - \eta_1\mathcal{H}\left[\pi_\theta\right] - \eta_2\mathcal{H}\left[\pi_\theta^{mask}\right] \Bigg\} \tag{2}
$$

where the $KL_{prcp}$ is implemented as $\mathbb{D}_{KL}[\pi_\theta||\pi_\theta^{mask}] = r_i^{prcp}(\theta) - \log r_i^{prcp}(\theta) - 1$ following Schulman (2020). $i$ indexes the $i$-th rollout response. The entropy values for the Double Entropy Loss are implemented as $\mathcal{H}[\pi_\theta] = \log \pi_\theta(o|q, I)$, $\mathcal{H}[\pi_\theta^{mask}] = \log \pi_\theta(o|q, I_{mask})$. And $\gamma$, $\eta_1$ and $\eta_2$ are hyperparameters used for loss weighting. Similarly, we derive the DAPO-version objective of PAPO (**PAPO$_D$**), as shown in **Appendix B**.

**Masking Strategy.** We investigate two strategies for creating the corrupted visual input $I_{mask}$ for the $KL_{prcp}$ loss: (1) *random masking* and (2) *semantic-aware masking*. For both strategies, we first set a target masking ratio (e.g., 60 %), which determines the fraction of patches to be masked. We adopt patch-based masking rather than pixel-level noise (e.g., adding Gaussian noise) because patch-based masking more effectively removes informative semantic content, whereas pixel-level noise typically preserves semantics even at high noise levels (see Figure 3 for a comparison).

In random masking, patches are selected uniformly. In semantic-aware masking, we leverage DINOv2 (Oquab et al., 2023), a self-supervised, pre-trained vision encoder, to identify salient patches: we aggregate its patch-level self-attention scores and select the top-scoring patches (see Appendix G for details). Empirically, we find that random masking yields better performance with negligible computational overhead (detailed in §5.2).

## 4 EXPERIMENTS

### 4.1 EXPERIMENTAL SETUP

We train all models on ViRL39K (Wang et al., 2025b) for 2 epochs using a learning rate of 1e-6. We perform direct RL training from Qwen2.5-VL-3B, 7B and Qwen3-VL-2B, comparing the standard

Table 1: **Performance (avg@8 acc %) comparison** of Qwen2.5-VL and Qwen3-VL models between GRPO, DAPO and PAPO on general and more vision-dependent multimodal reasoning tasks. MathVerse$_V$ refers to the vision-centric subset of MathVerse (Zhang et al., 2024). $\Delta^{\%}_{rel}$ indicates the averaged relative gain over the baseline for each task. We observe consistent improvements against both GRPO and DAPO, with gains approaching 8%-19%, especially on tasks with high vision-dependency. Training dynamics for these models are compared in Figure 4.

| Method | General Multimodal Reasoning | | | | | | | Vision-Dependent Multimodal Reasoning | | | | | | Overall | |
|---|---|---|---|---|---|---|---|---|---|---|---|---|---|---|---|
| | Geo3k | MathVista | We-Math | MMKI2 | MathVerse | AVG | $\Delta^{\%}_{rel}$ | LogicVista | Counting | MMMU-Pro | MathVerse$_V$ | AVG | $\Delta^{\%}_{rel}$ | AVG | $\Delta^{\%}_{rel}$ |
| **Qwen2.5-VL** | | | | | | | | | | | | | | | |
| GRPO-3B | 28.72 | 59.34 | 58.90 | 57.24 | 55.25 | 51.89 | – | 38.14 | 55.81 | 25.66 | 52.26 | 42.97 | – | 47.92 | – |
| **PAPO$_G$-3B** | **30.95** | **61.38** | **60.09** | **57.39** | **57.14** | **53.39** | ↑3.38 | **38.67** | **62.56** | **27.11** | **53.95** | **45.57** | ↑5.60 | **49.92** | ↑4.36 |
| GRPO-7B | 40.18 | 65.48 | **68.12** | 72.26 | 66.51 | 62.51 | – | 45.62 | 73.94 | 35.17 | 61.71 | 54.11 | – | 58.78 | – |
| **PAPO$_G$-7B** | **40.25** | **69.53** | 66.79 | **72.52** | **68.43** | **63.50** | ↑1.53 | **46.07** | **89.81** | **36.63** | **64.97** | **59.37** | ↑7.96 | **61.66** | ↑4.39 |
| DAPO-3B | 31.20 | 60.89 | 59.95 | **66.83** | 56.25 | 55.02 | – | 40.69 | 74.25 | 28.42 | 53.09 | 49.11 | – | 52.40 | – |
| **PAPO$_D$-3B** | **35.65** | **62.53** | **62.67** | 64.09 | **60.51** | **57.09** | ↑5.00 | **41.67** | **83.56** | **28.76** | **57.72** | **52.93** | ↑5.97 | **55.24** | ↑5.54 |
| DAPO-7B | 35.92 | 61.91 | 58.51 | 75.93 | 55.64 | 57.58 | – | 37.05 | 90.05 | 29.02 | 51.04 | 51.79 | – | 55.01 | – |
| **PAPO$_D$-7B** | **44.11** | **67.53** | **68.30** | **80.61** | **68.58** | **65.83** | ↑15.61 | **46.70** | **91.38** | **36.34** | **64.87** | **59.82** | ↑19.09 | **63.16** | ↑17.54 |
| **Qwen3-VL (thinking)** | | | | | | | | | | | | | | | |
| GRPO-2B | 39.29 | 53.58 | 57.12 | 47.71 | 47.98 | 49.13 | – | 29.84 | 80.13 | 20.51 | 45.41 | 43.97 | – | 46.84 | – |
| **PAPO$_G$-2B** | **41.08** | **56.08** | **59.17** | **48.57** | **51.89** | **51.36** | ↑4.52 | **32.83** | **80.63** | **23.42** | **50.05** | **46.73** | ↑6.27 | **49.30** | ↑5.25 |

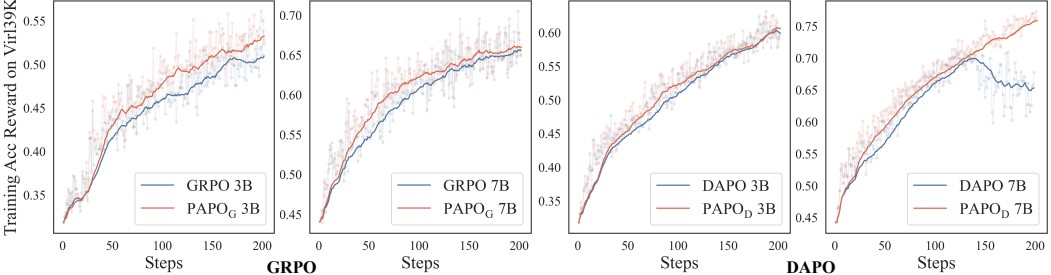

Figure 4: **Comparison of the training dynamics on the accuracy reward.** Solid lines indicate running averages with a stepping window size of 20. PAPO demonstrates consistently faster learning from the early stages on both GRPO and DAPO. Notably, DAPO-7B suffers from model collapse in the later stages, whereas PAPO$_D$ achieves continued improvements without collapse, highlighting the effectiveness of the proposed Double Entropy regularization. Further analysis on regularizing the DAPO baseline is presented in Appendix H.

GRPO and DAPO baselines with our proposed variants, PAPO$_G$ and PAPO$_D$. Note that GRPO uses a reference KL penalty, while DAPO removes it and employs dynamic sampling. Additional details on the hyperparameter configurations are provided in Appendix C.

## 4.2 EVALUATION

To systematically evaluate the effectiveness of PAPO, we conduct experiments and ablation studies on eight benchmarks that cover diverse multimodal reasoning problems, including: **(1) Math and Geometric Reasoning**: Geometry3K (Lu et al., 2021), MathVista (Lu et al., 2023), MathVerse (Zhang et al., 2024), and We-Math (Qiao et al., 2024); **(2) Multi-discipline Multimodal Reasoning**: MMMU-Pro (Yue et al., 2024); **(3) Logical Reasoning**: LogicVista (Xiao et al., 2024); **(4) Counting**: SuperClevr Counting (Li et al., 2023). All evaluation metrics are based on exact match against ground truth answer. We report average accurary @ 8 for all benchmarks with a inference temperature of 1.0. We omit datasets or instances with free-form answers that require LLM-as-a-judge evaluation.

**Analysis on Vision Dependency.** As also discussed in (Zhang et al., 2024; Yue et al., 2024), we observe that not all mainstream multimodal benchmarks are guaranteed to have visually dependent problems. That is, some reasoning tasks may rely heavily on textual content and do not require the visual content for deriving the answer. For example, Figure 6 exhibits VQA problems of different vision-dependency levels, highlighting the varying degrees of reliance on visual information in multimodal question answering. To this end, we conduct a manual screening of the included benchmarks and identify the following two categories: **(1) Vision-Dependent Multimodal Reasoning:** Benchmarks in which instances explicitly require proper interpretation of the visual input; **(2) General**

Table 2: **Results on integrating PAPO with modifications from other perspectives (e.g., Noisy-Rollout).** The base model is Qwen2.5-VL-3B. We find that NoisyRollout yields substantial gains on some datasets but shows inconsistent improvements across different tasks (reductions on 4/9 benchmarks). PAPO is compatible with NoisyRollout and delivers additional gains when combined.

| Method | General Multimodal Reasoning | | | | | Vision-Dependent Multimodal Reasoning | | | | Overall |
|---|---|---|---|---|---|---|---|---|---|---|
| | Geo3k | MathVista | We-Math | MMKI2 | MathVerse | LogicVista | Counting | MMMU-Pro | MathVerse$_V$ | AVG |
| GRPO | 28.72 | 59.34 | 58.90 | 57.24 | 55.25 | 38.14 | 55.81 | 25.66 | 52.26 | 47.92 |
| GRPO + NoisyRollout | 28.49 | 61.34 | 58.58 | 66.78 | **61.34** | 37.47 | 61.44 | 28.79 | 51.30 | 50.61 |
| **PAPO**$_G$ | **30.95** | **61.38** | 60.09 | 57.39 | 57.14 | 38.67 | 62.56 | 27.11 | **53.95** | 49.92 |
| **PAPO**$_G$ **+ NoisyRollout** | 30.66 | 58.83 | **60.60** | **66.95** | 56.21 | **39.51** | **71.38** | **29.47** | 53.36 | **51.89** |

**Multimodal Reasoning:** Benchmarks where instances may place weaker requirements on attending to the visual input when answering the question. More details are presented in Appendix D.

## 5    RESULTS

### 5.1    MAIN RESULTS

**PAPO consistently outperforms GRPO and DAPO for multimodal reasoning.**    We present the main results on both 3B and 7B base models in **Table 1**. The $\Delta_{rel}^{\%}$ denotes the average relative gain against the GRPO/DAPO baseline across all tasks. PAPO shows consistent overall improvements (4.4%-17.5%), with identical training dataset, rollout space and reward design, compared to the baselines. In **Figure 4**, we further present a comparison of training dynamics based on the accuracy rewards on ViRL39K. PAPO showcases faster learning even from the early steps. While DAPO-7B encounters model collapse in later stages, PAPO$_D$ continues to improve steadily, demonstrating the efficacy of the proposed Double Entropy regularization. In **Appendix H**, we further explore regularizing the DAPO baseline with our proposed entropy-based loss and demonstrate improvements over the stronger, regularized DAPO baseline. In **Appendix E**, we provide additional qualitative analysis of the attention patterns, illustrating how PAPO encourages stronger and more accurate attention to the image patches.

**More significant improvements on vision-dependent reasoning.**    The performance gains are more pronounced on the vision-dependent subset, leading to a relative gain of 8.0%-19.1%. This aligns with our expectation regarding the impact of Implicit Perception Loss, as it encourages visually dependent responses.

**Significantly reduced perception errors.**    We further conduct a comprehensive qualitative study on the error distribution, following the setup detailed in §2.2. **Figure 1** shows a before-and-after comparison of errors with GRPO and PAPO. We observe a significant reduction in perception errors, demonstrating the effectiveness of addressing the perception bottleneck in GRPO for multimodal reasoning. In **Appendix K**, we provide additional evaluations on OCR-Bench-v2 (Fu et al., 2024), which demonstrate the benefits of PAPO on fine-grained perception tasks. In **Appendix N**, we further analyze the perception errors and their outcomes under PAPO, investigating whether these errors are corrected or shift to other categories.

**Compatibility with other algorithmic modifications.**    As discussed in §1, prior work has focused on modifying the RLVR framework from the data, rollout, and reward perspectives. Since PAPO modifies only the optimization objective, it is in theory complementary to modifications from these other perspectives. To further examine this, we integrate PAPO with NoisyRollout (Liu et al., 2025a). The results are presented in Table 2. We summarize our findings as follows: (1) NoisyRollout yields inconsistent improvements across tasks, achieving large gains on some tasks but reductions on 4 out of 9 benchmarks. (2) Combining PAPO with NoisyRollout yields complementary gains, demonstrating strong compatibility with advances from other perspectives, such as the rollout stage.

**Robustness in GRPO with removed KL penalty.**    To conduct a more controlled investigation of the robustness of PAPO with KL$_{prcp}$ under the removal of the original KL penalty, we consider an additional GRPO variant where the reference KL is removed without introducing other modifications like in DAPO. **Table 8** (in Appendix F) shows that PAPO achieves overall improvements of 11.2% and 4.0% on the 3B and 7B models, respectively, outperforming the GRPO + Removed KL baselines.

**Robustness in low vision-dependent tasks.** Currently, PAPO applies the Implicit Perception Loss loss to all instances and all tokens, encouraging perceiving the visual inputs. Although this minimalist design works empirically well, we further investigate the model's robustness to strictly vision-independent scenarios. Specifically, we consider a text-only general reasoning benchmark MMLU-pro (Wang et al., 2024), and insert dummy visual inputs into the context (images containing pure random noise). If the model were to attend to these noisy visual tokens indiscriminately, we would expect a degradation in performance. The results are presented in Table 13 (Appendix). We find that PAPO still achieves competitive or stronger performance under this extreme setting, indicating good robustness in low-vision-dependency scenarios.

## 5.2 ABLATION ON KEY DESIGN CHOICES

**Impact of Masking Ratio and Strategy.** We investigate the most effective way to corrupt the original visual input to maximize the benefit of PAPO. In **Table 3**, we compare both masking strategies on $PAPO_G$, i.e., random masking vs. semantic-aware masking, and masking ratios, which control the percentage of patches to be masked. Implementation details for the masking strategies are provided in Appendix G. We find:

- Random masking empirically outperforms semantic-aware masking. We hypothesize that semantic-aware masking underperforms because, as illustrated in Figure 3, it tends to obscure entire salient regions, causing the model to attend to all objects indiscriminately instead of focusing on the most informative parts.
- Masking a sufficiently large portion of the image, e.g., 0.6 to 0.8, results in best performances. However, using a completely blackened image is not favorable, as it encourages the model to attend to the image regardless of its content. We also observe that complete blackening is more likely to cause $KL_{prcp}$ Hacking (detailed in §5.3).

Table 3: **Impact of masking strategy and ratio.** Performance comparison of $PAPO_G$ using different approaches for constructing $I_{mask}$. The base model is Qwen2.5-VL. Despite its simplicity, random masking empirically outperforms semantic-aware masking. A sufficiently large masking ratio (e.g., 0.6) yields stronger performance. See details in §5.2.

| Model | | General | | Vision | | Overall | |
|---|---|---|---|---|---|---|---|
| Size | Method | AVG | $\Delta_{rel}^{\%}$ | AVG | $\Delta_{rel}^{\%}$ | AVG | $\Delta_{rel}^{\%}$ |
| **GRPO Baselines** | | | | | | | |
| **3B** | GRPO | 51.89 | – | 42.97 | – | 47.92 | – |
| **7B** | GRPO | 62.51 | – | 54.11 | – | 58.78 | – |
| **Impact of Masking Strategy on PAPO** | | | | | | | |
| **3B** | **random** @0.6 | **52.53** | ↑ 1.73 | **45.17** | ↑ 4.52 | **49.26** | ↑ 2.97 |
| | semantic @0.6 | 52.13 | ↑ 0.34 | 43.78 | ↑ 1.88 | 48.42 | ↑ 1.02 |
| **7B** | **random** @0.6 | **63.56** | ↑ 1.91 | **57.49** | ↑ 5.37 | **60.86** | ↑ 3.55 |
| | semantic @0.6 | 63.39 | ↑ 1.48 | 56.83 | ↑ 3.89 | 60.47 | ↑ 2.55 |
| **Impact of Masking Ratio on PAPO** | | | | | | | |
| **3B** | random @**0.4** | 52.51 | ↑ 1.55 | 44.12 | ↑ 2.29 | 48.78 | ↑ 1.88 |
| | random @**0.6** | 52.53 | ↑ 1.73 | 45.17 | ↑ 4.52 | 49.26 | ↑ 2.97 |
| | random @**0.8** | **52.57** | ↑ 1.49 | 44.24 | ↑ 2.69 | 48.87 | ↑ 2.02 |
| | random @**1.0** | 52.13 | ↑ 0.71 | 43.98 | ↑ 2.31 | 48.51 | ↑ 1.42 |

**Impact of Implicit Perception Loss weighting.** We ablate on the choice of $\gamma$, which is the weighting coefficient of $KL_{prcp}$ as shown in Equation 2. **Table 4** presents the performance comparison based on $PAPO_G$-3B when varying $\gamma$ from 0.005 to 0.04. We summarize our findings as follows:

- A larger $\gamma$ under 0.02 tends to result in more pronounced improvements, especially on more visually-dependent tasks. Without additional regularization, setting $\gamma = 0.02$ for $PAPO_G$-3B models and $\gamma = 0.01$ for $PAPO_G$-7B models serves as a good default.
- $\gamma$ should not be set too large (e.g., 0.04), as it causes severe model collapse that cannot be regularized even with Double Entropy Loss. We also observe that larger models are more

Table 4: **Impact of $KL_{prcp}$ loss weighting.** Performance comparison on $PAPO_G$ with Qwen2.5-VL-3B using different values of $\gamma$. Increasing $\gamma$ up to 0.02 generally improves performance, while an excessively large $\gamma$, such as 0.04, leads to model collapse (see detailed discussion in §5.3). Larger models are also more sensitive to high $\gamma$ as shown in Figure 12.

| Method | General | | Vision | | Overall | |
|---|---|---|---|---|---|---|
| | AVG | $\Delta_{rel}^{\%}$ | AVG | $\Delta_{rel}^{\%}$ | AVG | $\Delta_{rel}^{\%}$ |
| GRPO | 51.89 | – | 42.97 | – | 47.92 | – |
| **PAPO @0.005** | 52.40 | ↑ 1.19 | 43.73 | ↑ 1.92 | 48.55 | ↑ 1.51 |
| **PAPO @0.01** | 52.53 | ↑ 1.73 | 45.17 | ↑ 4.52 | 49.26 | ↑ 2.97 |
| **PAPO @0.02** | **53.39** | ↑ 3.38 | **45.57** | ↑ 5.60 | **49.92** | ↑ 4.36 |
| **PAPO @0.04 (collapsed)** | 31.24 | ↓ 43.15 | 38.31 | ↓ 14.09 | 34.38 | ↓ 28.46 |

be regularized even with Double Entropy Loss. We also observe that larger models are more

sensitive to higher $\gamma$ values and require earlier regularization as shown in **Figure 12** in the Appendix. We further discuss the impact of $\gamma$ on $\text{KL}_{\text{prcp}}$ Hacking in §5.3.

- In settings without a reference KL penalty (including $\text{PAPO}_D$), $\gamma$ needs to be set more conservatively (e.g., 0.01), and Double Entropy Loss is indispensable (see **Figure 10** in the Appendix).

## 5.3 DEEP DIVE ON TRAINING STABILITY

In this section, we aim to gain a deeper understanding of $\text{KL}_{\text{prcp}}$ Hacking, a unique failure mode where the model over-optimizes the Implicit Perception Loss. We present our findings on: **(1)** the model's generation behavior after collapsing; **(2)** early signs indicating a model collapse; **(3)** the most influential factors contributing to the hacking; and **(4)** regularization approaches to prevent or delay its occurrence.

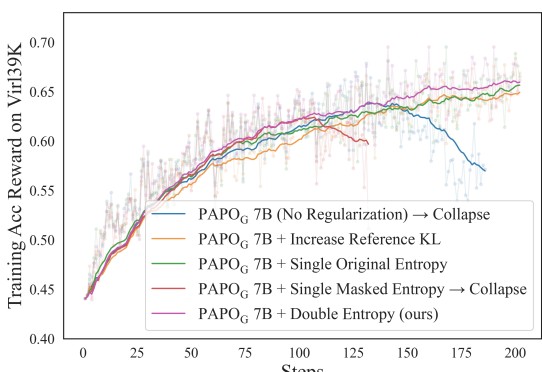

Figure 5: **Comparison of different regularization strategies.** All strategies are applied to the same collapsing baseline, $\text{PAPO}_G$ ($\gamma = 0.02$, no regularization). Among the four methods described in the main text, three successfully prevent the collapse entirely, while adding Single Masked Entropy only delays it. The proposed Double Entropy Loss demonstrates the best training dynamics and prevents the collapse. Detailed results are compared in Table 5.

**Collapsing behavior.** We first examine how the model behaves after collapsing in terms of its generation. We manually examine the generated tokens of a collapsed model, a $\text{PAPO}_G$-7B with $\gamma = 0.02$ (without regularization), and a non-collapsing model, GRPO-7B. We observe a notably abnormal generation behavior, namely, a tendency to generate entirely unrelated tokens during reasoning (see **Figure 13** in the Appendix). We quantitatively verify this by leveraging GPT-4.1-mini (OpenAI, 2025) as a judge to score the relatedness of the model's response. As presented in the bottom left of Figure 13, the collapsed model shows significantly lower relatedness. See **Appendix M** for more details on this experiment.

**Early signs of $\text{KL}_{\text{prcp}}$ Hacking.** When the hacking occurs, the model exhibits a drastic decrease in $\text{KL}_{\text{prcp}}$, accompanied by collapsing training rewards. Meanwhile, the Clipping Ratio-High increases, indicating a growing proportion of tokens undergoing policy gradient updates beyond the clipping threshold, which serves as an early sign of collapse. We also observe an interesting pattern: the entropy loss for both the original $\pi_\theta$ and the corrupted policy $\pi_\theta^{\text{mask}}$ increases as the collapse unfolds. This observation inspires our regularization strategies, which are detailed later in this section. See **Figure 11** in the Appendix for the metric dynamics.

**Influential factors towards $\text{KL}_{\text{prcp}}$ Hacking.** We summarize three main factors that make the model more prone to $\text{KL}_{\text{prcp}}$ Hacking: **(1) Model size:** Larger models tend to be more sensitive to hacking under the same configuration. For example, setting $\gamma = 0.02$ causes collapse on 7B models but not on 3B models. (See Figure 12 a.) **(2) Loss weighting:** A higher $\text{KL}_{\text{prcp}}$ weighting, such as 0.04, is more likely to lead to collapse. (See Figure 12 b.) **(3) Masking ratio:** Using an extreme masking ratio, e.g., 1.0, leads to a faster collapse. (See Figure 12 c.)

Table 5: **Performance comparison between the three regularization methods** that successfully prevent model collapse, as shown in Figure 5. The base model is Qwen2.5-VL-7B. Among these methods, Double Entropy Loss achieves the best overall improvement of 4.4%.

| Method | General | | Vision | | Overall | |
|---|---|---|---|---|---|---|
| | **AVG** | $\Delta_{rel}^\%$ | **AVG** | $\Delta_{rel}^\%$ | **AVG** | $\Delta_{rel}^\%$ |
| GRPO | 62.51 | – | 54.11 | – | 58.78 | – |
| $\text{PAPO}_G$ w/ Inc $\text{KL}_{\text{ref}}$ | 63.14 | ↑ 1.12 | 57.03 | ↑ 3.99 | 60.42 | ↑ 2.40 |
| $\text{PAPO}_G$ w/ Single Ent | 63.34 | ↑ 1.53 | 58.36 | ↑ 5.96 | 61.12 | ↑ 3.50 |
| $\text{PAPO}_G$ w/ **Double Ent** | **63.50** | ↑ 1.53 | **59.37** | ↑ 7.96 | **61.66** | ↑ 4.39 |

**Regularization approaches for preventing $\text{KL}_{\text{prcp}}$ Hacking.** Inspired by the aforementioned findings, we investigate four different approaches to prevent the collapse: **(1)** Increasing the KL penalty against the reference model; **(2)** Adding a single entropy loss on the original policy sequence $\pi_\theta$; **(3)** Adding a single entropy loss on the corrupted policy sequence $\pi_\theta^{\text{mask}}$; and **(4)** Adding a **Double**

**Entropy Loss** on both $\pi_\theta$ and $\pi_\theta^{\text{mask}}$. See **Figure 5** in the Appendix for the training dynamics of the four approaches. Among them, (1)(2)(4) successfully prevent the collapse from happening.

We further examine their evaluation performance, as shown in **Table 5**. We find that adding entropy-based loss on the original sequence is essential for regularizing training stability. Overall, Double Entropy Loss achieves the most significant performance gain while also successfully avoiding $\text{KL}_{\text{prcp}}$ Hacking. In **Figure 9** in the Appendix, we also show that adding entropy loss alone fails to prevent DAPO from collapsing, whereas $\text{PAPO}_D$ stabilizes training and yields significant improvements.

## 6 RELATED WORK

Prior work on multimodal RLVR have primarily focused on enhancing three components of the original GRPO framework: **Data**, **Rollout**, and **Reward**, while leaving the core optimization objectives largely untouched. We offer a comprehensive categorization and comparison in Table 15 in the Appendix.

**Data-Centric Approaches.** Earlier efforts such as R1-V (Chen et al., 2025; Huang et al., 2025; Meng et al., 2025a) distill chain-of-thought (CoT) data from strong textual reasoning models like Deepseek-R1 (Guo et al., 2025) and explore directly applying R1-style training pipelines to multimodal tasks, demonstrating initial promise in generalization. Recent work such as MoDoMoDo (Liang et al., 2025; Li et al., 2025a) explores more sophisticated sample selection mechanisms to improve data quality.

**Rollout Improvements.** NoisyRollout (Liu et al., 2025a) and R1-ShareVL (Yao et al., 2025) show the benefits of diversifying the rollout space using responses generated from moderately augmented visual inputs. VL-Rethinker (Wang et al., 2025b) and Skywork R1V2 (Wang et al., 2025d) adopt a Selective Sample Replay mechanism to mitigate the prevalent issue of vanishing advantages.

**Reward Enhancements.** Several approaches (Ma et al., 2025; Liu et al., 2025b; Fan et al., 2025) incorporate grounding-related metrics, such as IoU for bounding boxes. Visionary-R1 (Xia et al., 2025) introduces captioning-based rewards, prompting the model to generate detailed textual descriptions of visual input before reasoning. While initially promising, this approach enforces a separation between perception and reasoning, which can be suboptimal for capturing low-level visual details.

**Perception as Tool-Using.** Another line of emerging work takes a different view on improving perception in multimodal reasoning, relying on tool use for perception. Recent efforts such as DeepEyes (Zheng et al., 2025), ACTIVE-O3 (Zhu et al., 2025b), and Pixel Reasoner (Su et al., 2025) enhance perception by incentivizing the LMM to perform visual operations, such as zooming in. However, these methods do not directly improve the native perception capabilities of the multimodal models.

Consequently, we find that the prevailing assumption: multimodal reasoning can be effectively addressed solely through data- and reward-level modifications to text-based RL, is inherently limiting. Our work challenges this paradigm by demonstrating that incentivizing visually grounded reasoning requires deeper integration into the core **optimization objective**, rather than treating vision as a secondary modality addressed through auxiliary adjustments.

## 7 CONCLUSION AND LIMITATIONS

In this paper, we present PAPO, a novel policy gradient algorithm that encourages the reasoning steps in Large Multimodal Models (LMMs) to be internally grounded in visual inputs. PAPO requires no additional annotations, no reliance on stronger teacher models, and no expensive neural reward models, making it a direct drop-in replacement to GRPO or DAPO. Despite its simplicity, PAPO significantly improves complex visual reasoning. One limitation of our current work is that we have not yet explored scaling to larger model sizes or evaluating compatibility with other model families, such as the InternVL (Zhu et al., 2025a) series. Additionally, while PAPO introduces only moderate computational overhead (see Appendix O), we have not focused on optimizing training efficiency, which remains an important direction for future research. Another promising avenue for future work is investigating a finer-grained selection of tokens on which to apply the PAPO objective, rather than applying it uniformly to all tokens.

## 8 REPRODUCIBILITY STATEMENT

We include an anonymous source code in the supplementary material, containing training and evaluation instructions for reproducing the results in this paper. Implementation details of the datasets and models are provided in §C. The prompt used for the LLM-as-a-judge evaluation in Figure 13 is presented in Figure 14.

## 9 ETHICS STATEMENT

This work focuses on fundamental research in reinforcement learning and multimodal reasoning. Our methods are developed and evaluated entirely on publicly available benchmarks without involving human subjects, sensitive personal data, or private information. The proposed algorithm, PAPO, is designed as a general optimization technique and does not raise concerns regarding fairness, bias, discrimination, privacy, or security. We believe that our study poses no foreseeable ethical risks and fully complies with research integrity standards.

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

APPENDIX

## A USE OF LARGE LANGUAGE MODELS STATEMENT

Large language models, such as ChatGPT, are used exclusively for grammar checking during the writing process. They are not used for research ideation. In addition, LLM-based coding assistants, such as Copilot, are employed to support code implementation.

## B PAPO$_D$ OBJECTIVE

Similar to PAPO$_G$, we extend DAPO (Yu et al., 2025) with our perception-aware supervision signal to implement PAPO$_D$. The complete PAPO$_D$ objective is shown as follows:

$$
\mathcal{J}_{\text{PAPO}_D}(\theta) = \mathbb{E}_{[\{o_i\}_{i=1}^G \sim \pi_{\theta_{old}}(O|q,I)]} \frac{1}{\sum_{i=1}^G |o_i|} \sum_{i=1}^G \sum_{t=1}^{|o_i|} \Big\{
$$

$$
\min\Big[r_{i,t}(\theta)\hat{A}_{i,t}, \text{clip}\left(r_{i,t}(\theta), 1-\epsilon_l, 1+\epsilon_h\right)\hat{A}_{i,t}\Big] {\color{blue}+\gamma\mathbb{D}_{\text{KL}}[\pi_\theta||\pi_\theta^{\text{mask}}] - \eta_1\mathcal{H}[\pi_\theta] - \eta_2\mathcal{H}[\pi_\theta^{mask}]}\Big\}
$$

$$
\text{with } 0 < |\{o_i \mid \texttt{is\_equivalent}(a, o_i)\}| < G \tag{3}
$$

where $\gamma$ is the weighting coefficient for KL$_{\text{prcp}}$, $\eta_1$ and $\eta_2$ are the weighting coefficients for the Double Entropy Loss terms, and $r_i^{\text{prcp}}(\theta) = \frac{\pi_\theta(o_i|q,I)}{\pi_\theta(o_i|q,I_{\text{mask}})}$ quantifies the model's reliance on visual information. We implement $KL_{\text{prcp}}$ using the same approximation method as in PAPO$_G$. Note that in PAPO$_D$, the reference KL penalty ($\mathbb{D}_{KL}[\pi_\theta||\pi_{ref}]$) is removed as in DAPO. In addition, the dynamic sampling strategy is enabled to prevent sampling instances with either all-correct or all-wrong rollouts. The maximum number of retries for dynamic sampling is set to 20 for both the DAPO baseline and PAPO$_D$.

## C IMPLEMENTATION DETAILS

**Dataset.** We use ViRL39K (Wang et al., 2025b), a diverse collection of 38.8K multimodal reasoning QA pairs covering problems in math, STEM, charts, and social topics. Note that the training data includes only input queries and final answers, without intermediate chain-of-thought (CoT) annotations.

**Models.** We employ the Qwen2.5-VL-3B (Qwen Team, 2024a), Qwen2.5-VL-7B (Qwen Team, 2024b) and Qwen3-VL-2B-thinking (Qwen Team, 2024c) as our base models. By default, the base model refers to Qwen2.5-VL. We consider the following main model variants and default configurations:

- **GRPO**: GRPO (Shao et al., 2024) baseline with clipping factors set to $\epsilon_l = 0.2$, $\epsilon_h = 0.3$, and reference KL penalty coefficient set to $\beta = 0.01$.
- **DAPO**: DAPO (Yu et al., 2025) baseline with clipping factors set to $\epsilon_l = 0.2$, $\epsilon_h = 0.28$, reference KL removed, token-level loss averaging enabled, and dynamic sampling with a maximum of 20 retries.
- **PAPO$_G$**: PAPO instantiated from GRPO.
- **PAPO$_D$**: PAPO instantiated from DAPO.

Table 6 summarizes the hyperparameter configurations for the best performing model variants as shown in Table 1. Detailed ablations and analysis are presented in §5.2 and §5.3. For PAPO models, we use random masking with a default masking ratio of 0.6. Double Entropy Loss is essential for settings with higher $\gamma$ on 7B models, and for configurations without the reference KL penalty (i.e., PAPO$_D$), due to the inherently weaker regularization against deviation from the base model.

**Training.** We conduct RLVR on all model variants using the following typical response format, where reasoning steps are enclosed in `<think></think>` and the final answer is enclosed in `\boxed{}`. Each model is trained for 2 epochs on ViRL39K (Wang et al., 2025b) with a learning rate of 1e-6 and weight decay of 1e-2. We use 2 and 4 NVIDIA H100 80G GPUs for 3B and 7B

| Model | $\gamma$ | $\eta_1, \eta_2$ | Mask Ratio | $\beta$ | $\varepsilon_l, \varepsilon_h$ |
|---|---|---|---|---|---|
| Qwen2.5-VL GRPO-3B | - | - | - | 0.01 | 0.2, 0.3 |
| Qwen2.5-VL GRPO-7B | - | - | - | 0.01 | 0.2, 0.3 |
| Qwen3-VL GRPO-2B | - | - | - | 0.01 | 0.2, 0.3 |
| Qwen2.5-VL PAPO$_G$-3B | 0.02 | - | 0.6 | 0.01 | 0.2, 0.3 |
| Qwen2.5-VL PAPO$_G$-7B | 0.02 | 0.05 | 0.6 | 0.01 | 0.2, 0.3 |
| Qwen3-VL PAPO$_G$-2B | 0.01 | - | 0.6 | 0.01 | 0.2, 0.3 |
| Qwen2.5-VL DAPO-3B | - | - | - | - | 0.2, 0.28 |
| Qwen2.5-VL DAPO-7B | - | - | - | - | 0.2, 0.28 |
| Qwen2.5-VL PAPO$_D$-3B | 0.01 | 0.03 | 0.6 | - | 0.2, 0.28 |
| Qwen2.5-VL PAPO$_D$-7B | 0.01 | 0.03 | 0.6 | - | 0.2, 0.28 |

Table 6: **Hyperparameter configurations** for models in Table 1.

experiments respectively. We use a rollout batchsize of 384, and generate $n = 5$ responses per prompt. More details on training configuration can be found in the code supplementary.

## D    VISION-DEPENDENCY ANALYSIS

We observe that current multimodal reasoning benchmarks exhibit varying degrees of reliance on visual information, ranging from questions answerable through textual cues alone to those requiring comprehensive visual analysis. To systematically characterize this spectrum, we propose a taxonomy of vision dependency levels and analyze their distribution across mainstream multimodal reasoning datasets, including ViRL39K (Wang et al., 2025b), Geometry3K (Lu et al., 2021), MathVista (Lu et al., 2023), MathVerse (Zhang et al., 2024), LogitVista (Xiao et al., 2024), MMK12 (Meng et al., 2025b), We-Math (Qiao et al., 2024), SuperClevr-Counting (Li et al., 2023), and MMMU-Pro (Yue et al., 2024). Specifically, we categorize vision dependency into three levels based on the significance distribution of critical information between textual questions and visual contents:

Table 7: **Analysis of vision-dependency** across nine multimodal reasoning datasets.

| Dataset | Low | Medium | High |
|---|---|---|---|
| **Training Dataset** | | | |
| ViRL39K (Wang et al., 2025b) | ✗ | ✓ | ✓ |
| **General Reasoning** | | | |
| Geo3K (Lu et al., 2021) | ✓ | ✓ | ✗ |
| LogicVista (Xiao et al., 2024) | ✗ | ✗ | ✓ |
| MathVerse (Zhang et al., 2024) | ✓ | ✓ | ✓ |
| MathVista (Lu et al., 2023) | ✓ | ✓ | ✓ |
| MMK12 (Meng et al., 2025b) | ✓ | ✓ | ✓ |
| We-Math (Qiao et al., 2024) | ✓ | ✓ | ✗ |
| **Vision-Dependent Reasoning** | | | |
| Counting (Li et al., 2023) | ✗ | ✗ | ✓ |
| MathVerse$_V$ (Zhang et al., 2024) | ✗ | ✓ | ✓ |
| MMMU-Pro (Yue et al., 2024) | ✗ | ✗ | ✓ |

- **Low:** In low vision-dependency tasks, the questions typically embed substantial visual information within the textual input itself, such as specifying the length of an important triangle side, thereby reducing the model's reliance on visual processing.

- **Medium:** Medium-level vision-dependency tasks provide partial contextual information textually while requiring the model to perceive and extract complementary visual features from the image.

- **High:** High vision-dependency tasks contain minimal or no visual information in the textual input, requiring the model to derive answers entirely through visual reasoning.

We manually examine the data instances and dataset construction pipelines for each benchmark and summarize our findings in Table 7. In Figure 6, we further illustrate representative examples from each category, demonstrating how the practical manifestation of these dependency levels directly correlates with the perception challenges we observe in current multimodal reasoning models. Notably, our experiments show that PAPO's improvements are most pronounced (8.0%) on high vision-dependency tasks, unveiling that learning perception-aware policies is essential for robust multimodal reasoning.

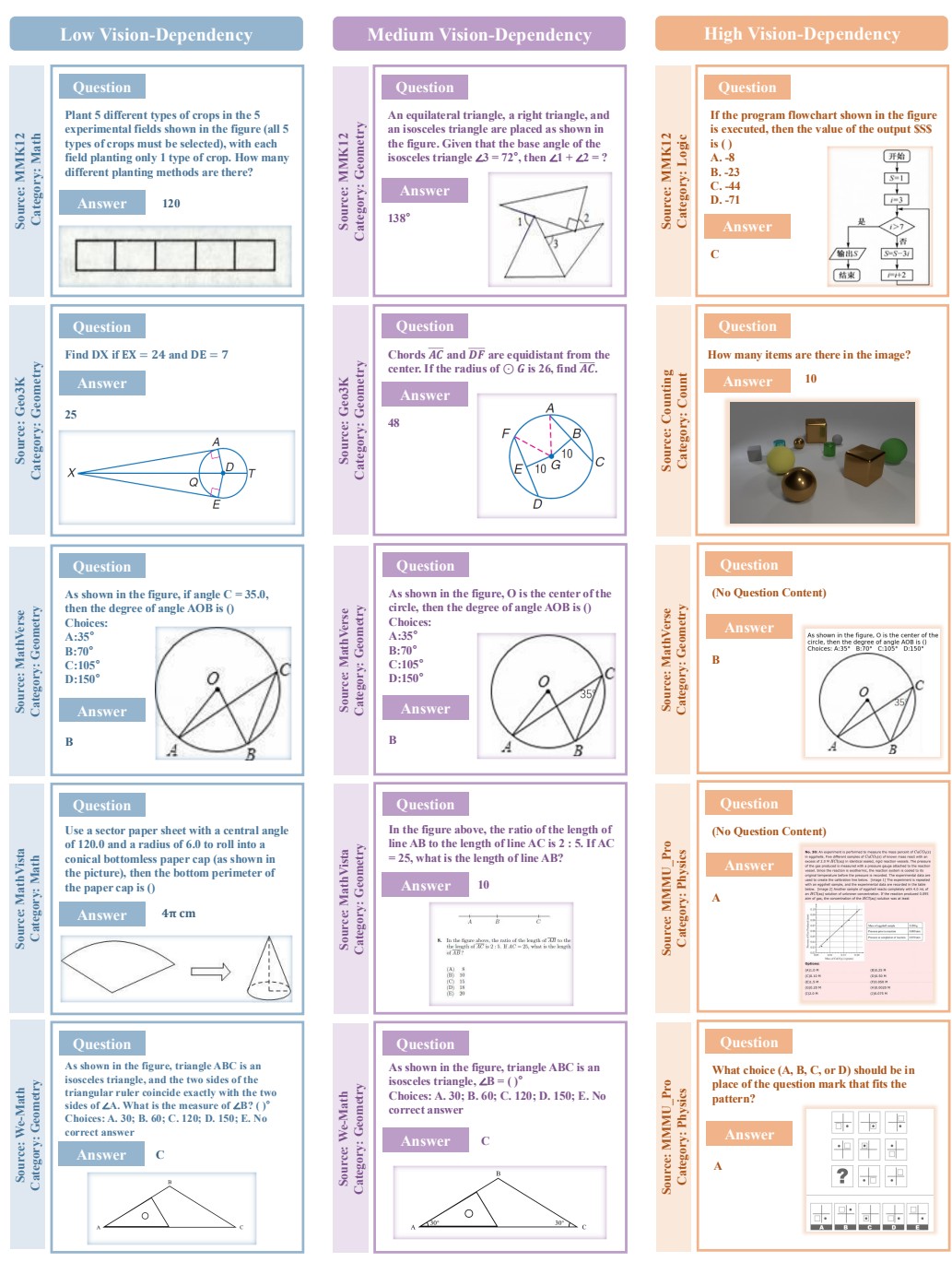

Figure 6: Illustrative examples of different levels of vision-dependency.

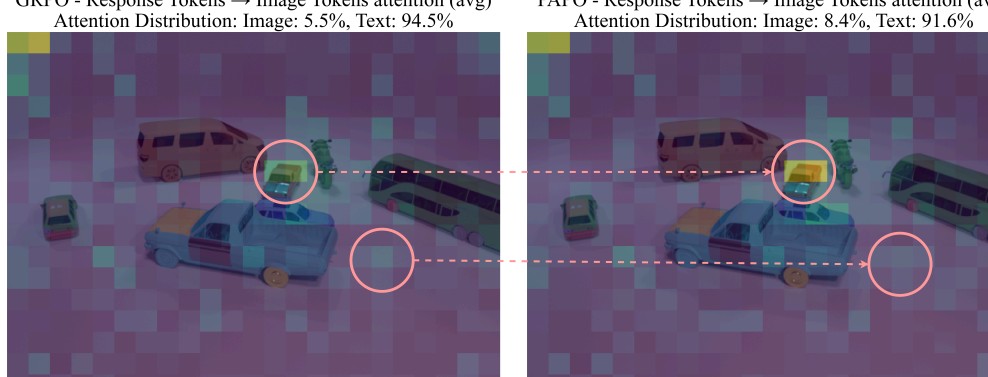

GRPO - Response Tokens → Image Tokens attention (avg)
Attention Distribution: Image: 5.5%, Text: 94.5%

PAPO - Response Tokens → Image Tokens attention (avg)
Attention Distribution: Image: 8.4%, Text: 91.6%

Figure 7: **Qualitative analysis of attention patterns** comparing the GRPO baseline and PAPO$_G$. We find that (1) PAPO encourages more attention on image tokens versus text tokens, and (2) PAPO places stronger attention on salient objects while reducing attention on empty regions (as highlighted in the circles).

Table 8: **Controlled experiments on reference KL removal.** For PAPO$_G$, we add a Double Entropy Loss with a coefficient of 0.03 for both 3B and 7B models. We find that PAPO$_G$ is highly compatible with this setting, achieving further improvements with an average relative gain of 11.2% and 4.0%. Improvements against the GRPO + No KL$_{ref}$ is more pronounced on more vision-dependent tasks, highlighting stronger perception capabilities for reasoning.

| Method | General | | Vision | | Overall | |
|---|---|---|---|---|---|---|
| | **AVG** | $\Delta^\%_{rel}$ | **AVG** | $\Delta^\%_{rel}$ | **AVG** | $\Delta^\%_{rel}$ |
| **3B** | | | | | | |
| GRPO | 51.89 | – | 42.97 | – | 47.92 | – |
| GRPO + No KL$_{ref}$ | 53.96 | ↑ 4.75 | 45.46 | ↑ 5.37 | 50.18 | ↑ 5.03 |
| **PAPO$_G$ + No KL$_{ref}$** | **56.21** | ↑ 9.26 | **49.33** | ↑ 13.60 | **53.15** | ↑ 11.19 |
| **7B** | | | | | | |
| GRPO | 62.51 | – | 54.11 | – | 58.78 | – |
| GRPO + No KL$_{ref}$ | **63.99** | ↑ 2.05 | 57.94 | ↑ 5.36 | 61.30 | ↑ 3.53 |
| **PAPO$_G$ + No KL$_{ref}$** | 63.31 | ↑ 1.15 | **59.18** | ↑ 7.54 | **61.47** | ↑ 3.99 |

# E   ADDITIONAL ANALYSIS ON ATTENTION PATTERN

We visualize the attention heatmap from the response tokens to the input image tokens. In Figure 7, we present the attention patterns of GRPO and PAPO, along with the overall attention distribution across image and text tokens in the inputs. We observe that (1) PAPO encourages more attention to be allocated to the input image tokens overall, and (2) PAPO directs attention more precisely toward salient objects while suppressing attention to empty or uninformative regions.

# F   CONTROLLED EXPERIMENTS ON REFERENCE KL REMOVAL WITH GRPO

To have a further controlled investigation on the robustness of PAPO with KL$_{prcp}$ under the removal of the original KL penalty, we consider an additional GRPO variant in which we remove the reference KL without introducing other modifications as in DAPO. Under this setting, we remove the $\mathbb{D}_{KL}[\pi_\theta||\pi_{ref}]$ term (referred to as $KL_{ref}$) from the GRPO and PAPO$_G$ training objective (Eq 2).

In Table 8, we compare GRPO + No KL$_{ref}$ with PAPO$_G$ + No KL$_{ref}$, using $\gamma = 0.01$ and Double Entropy Loss with $\eta_1 = \eta_2 = 0.03$. We observe that PAPO$_G$ performs well in this setting, achieving overall improvements of 11.2% and 4.0% for the 3B and 7B models, respectively. Its superiority

over the GRPO + No $KL_{ref}$ baseline is particularly evident on vision-dependent tasks, with gains of 13.6% and 7.5%, indicating enhanced perception capabilities for reasoning.

# G    MASKING STRATEGIES

We extend our elaboration on our two masking strategies for creating corrupted visual inputs $I_{\text{mask}}$ used in the Implicit Perception Loss $KL_{\text{prcp}}$. We show an illustration of different masking strategies in Figure 3. As visualized in Figure 3, we observe that patch-based masking removes informative semantic contents more effectively while pixel-level noises typically preserve semantics even at high noise levels. We detail the two patch-level masking strategies as follows.

## G.1    RANDOM MASKING

Random masking is a simple and computationally efficient approach for generating $I_{\text{mask}}$. Given an input image $I$ and a blackening probability $p \in [0, 1]$, we traverse the image in a grid pattern with patch size $s \times s$ pixels ($s = 14$ by default). For each patch location, we generate an independent random variable $r \sim \texttt{Uniform}(0, 1)$ and mask the patch if $r < p$.

This random masking implementation ensures that each patch has the probability $p$ of being masked, independent of other patches. The expected fraction of masked patches is $p$, with minimal computational overhead.

## G.2    SEMANTIC-AWARE MASKING

Semantic-aware masking aims to preferentially mask patches that contain more semantically important visual information. This approach leverages a pre-trained vision encoder to identify salient regions before applying masking. Our implementation uses DINOv2 (Oquab et al., 2023) as the vision encoder for its strong self-supervised representation learning capabilities.

**Attention-Based Saliency Computation.**    Given an input image $I$, we first process the image to obtain patch-level attention maps. For a model with $L$ layers and $H$ attention heads, this yields attention matrices $\mathbf{A}^{(l)} \in \mathbb{R}^{H \times N \times N}$ for each layer $l \in 1, 2, \ldots, L$, where $N$ is the total number of patches. As middle layers often capture more meaningful semantic relationships, we employ $6, 7, 8, 9$ layers to aggregate patch-level self-attention scores for saliency computation.

Specifically, for each selected layer $l$, aggregate attention across heads using mean pooling:

$$\overline{\mathbf{A}}^{(l)} = \frac{1}{H} \sum_{h=1}^{H} \mathbf{A}_h^{(l)} \tag{4}$$

The saliency score is computed for each patch $i$ by summing the attention it receives from all other patches:

$$s_i^{(l)} = \sum_{j=2}^{N} \overline{\mathbf{A}}_{j,i+1}^{(l)} \tag{5}$$

where the $+1$ offset accounts for the first $\texttt{CLS}$ token.

Saliency scores are averaged across selected layers:

$$s_i = \frac{1}{|L|} \sum_{l \in L} s_i^{(l)} \tag{6}$$

where $L = 6, 7, 8, 9$ is the set of selected layers.

**Saliency-Based Patch Selection.**    With computed saliency scores for all image patches, we select patches for masking through:

- **Ranking.** Sort patches in descending order of saliency scores to identify the most semantically important regions.

Table 9: **Normalized entropy of the attention distribution** (from the response tokens to the input image tokens), comparing PAPO trained with random masking and semantic masking.

| Model (Qwen2.5-VL) | Mean - Geo3K | Median - Geo3K | Mean - Counting | Median - Counting |
|---|---|---|---|---|
| 3B PAPO-G w/ Random Masking | 0.9520 | 0.9522 | 0.9351 | 0.9362 |
| 3B PAPO-G w/ Semantic Masking | 0.9522 | 0.9517 | 0.9331 | 0.9342 |
| 7B PAPO-G w/ Random Masking | 0.9457 | 0.9472 | 0.9525 | 0.9530 |
| 7B PAPO-G w/ Semantic Masking | 0.9401 | 0.9403 | 0.9513 | 0.9512 |

- **Top-k Selection.** Given masking ratio $p$, select the top $k = \lfloor p \times (N-1) \rfloor$ patches with highest saliency scores for masking.

- **Patch Masking.** Apply the same zero-out operation as in random masking to the selected high-saliency patches.

### G.3 FURTHER ANALYSIS OF MASKING STRATEGIES

In Table 3, we empirically observed that random masking performs better than semantic-aware masking. We believe the reason is that random masking naturally strikes a balance between masking out enough informative content while avoiding the removal of an entire "block" of salient information. A similar phenomenon has been observed in prior work on masked autoencoders (He et al., 2022) and masked visual modeling (Xie et al., 2022; Fu et al., 2023), where random masking is found to be competitive with, or even superior to, block-wise and attention-based masking strategies.

To further investigate what makes random masking performant, we conduct two additional analysis: (1) analyzing the attention patterns of PAPO models trained with random masking versus semantic masking, and (2) examining the error distribution in cases where the random-masking model succeeds but the semantic-masking model fails.

**Attention Pattern Analysis.** The goal of this experiment is to investigate whether the masking strategy leads to different attention behaviors after training. We extract the average attention weight from response tokens to input image tokens, and then compute the normalized entropy of the attention distribution across the image tokens. The normalized entropy is defined as $\text{Ent}(P)/\text{Ent}(P_{uni})$, where $P_{uni}$ is the uniform distribution over image patches; this normalization accounts for different numbers of image tokens. A higher entropy indicates a more scattered attention distribution, whereas a lower entropy indicates a more centralized one. We conduct this experiment on two datasets, Geo3K and Counting, sampling 100 instances per dataset.

As shown in Table 9, we find that the overall attention patterns do not differ significantly between the two masking strategies. However, the random-masking model generally exhibits slightly higher entropy, indicating a more scattered attention distribution. This observation aligns with our hypothesis that semantic masking encourages the model to focus more heavily on a centralized salient region.

**Error Analysis.** We also investigate what types of errors cause the semantic-masking model to underperform relative to the random-masking model. We sample 30 instances per dataset from Geo3K, Counting, and MathVerse where the random-masking model is correct but the semantic-masking model fails, and manually categorize the error types. As presented in Figure 8, we find that the predominant error source is still perception-related errors, suggesting that the primary advantage of random-masking over semantic-masking lies in improved perception ability.

## H   ADDITIONAL EXPERIMENTS ON REGULARIZING DAPO BASELINE

As shown in Figure 4, we observe model collapsing on the DAPO-7B baseline. In this section, we investigate further regularizing this baseline with an entropy loss and compare with PAPO$_D$.

Unlike PAPO that uses both $\pi_\theta$ and $\pi_\theta^{\text{mask}}$, DAPO baseline operates with a single policy $\pi_\theta$. Accordingly, we explore the effects of adding an entropy loss term to the DAPO objective:

$$\mathcal{J}_{\text{DAPO+Ent}}(\theta) = \mathcal{J}_{\text{DAPO}}(\theta) - \eta\mathcal{H}[\pi_\theta] \tag{7}$$

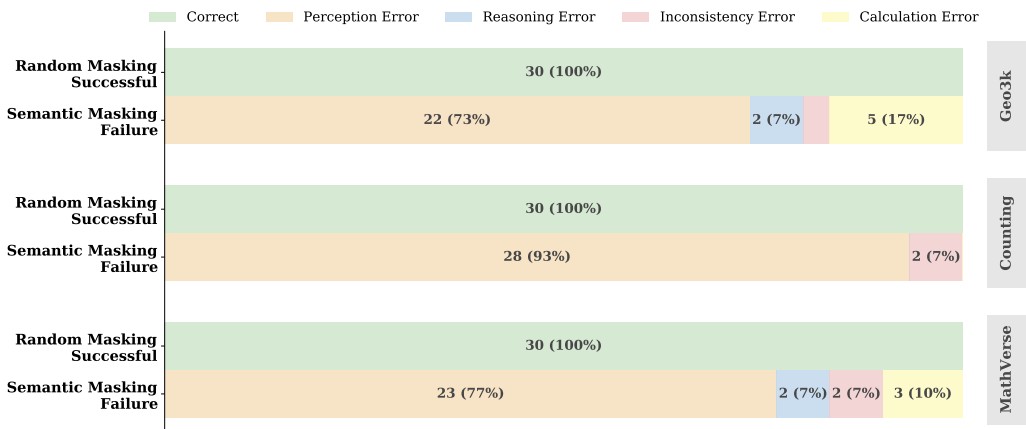

Figure 8: **Error categorization** for cases where the semantic-masking model fails but the random-masking model succeeds. We find that the most common reason for underperformance is poorer perception.

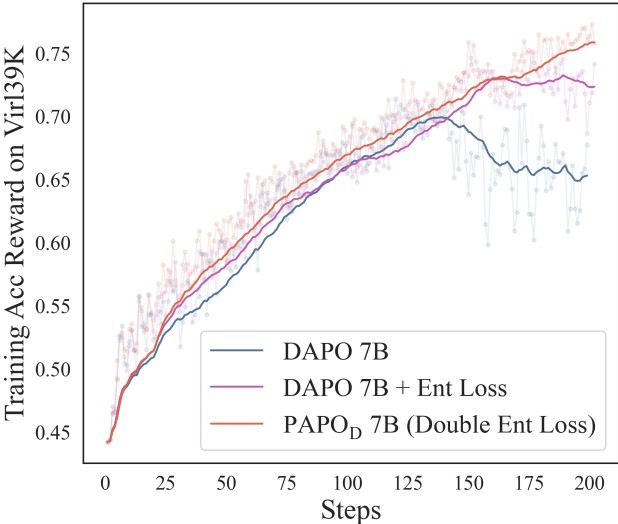

Figure 9: **Training dynamics of DAPO baseline with entropy loss.** Adding entropy loss to the DAPO-7B baseline effectively delays the collapse. PAPO$_D$ with Double Entropy Loss maintains more stable training and achieves superior performance. The comparison of benchmark performance is presented in Table 10.

where $\mathcal{H}$ is computed as the log probabilities of the response tokens, and $\eta$ is set to the same as $\eta_1 = 0.03$ as in PAPO$_D$. Figure 9 and Table 10 present the training dynamics and evaluation performance of the regularized DAPO baseline, compared with the original DAPO and PAPO$_D$. Our main findings are as follows:

- Adding single entropy loss to DAPO successfully delays collapse and improves end-task performance, but the regularization does not fully prevent collapse.

- PAPO$_D$ with Double Entropy Loss consistently outperforms both DAPO variants and completely prevents collapse throughout training.

Table 10: **Performance of DAPO baseline with entropy loss**. While adding entropy loss effectively reduces collapsing on DAPO-7B, $\text{PAPO}_D$ with Double Entropy Loss consistently outperforms this regularized DAPO baseline.

| Method | General | | Vision | | Overall | |
|---|---|---|---|---|---|---|
| | **AVG** | $\Delta_{rel}^{\%}$ | **AVG** | $\Delta_{rel}^{\%}$ | **AVG** | $\Delta_{rel}^{\%}$ |
| DAPO-7B | 57.58 | – | 51.79 | – | 55.01 | – |
| DAPO-7B + Ent Loss | 64.77 | ↑ 13.52 | 59.14 | ↑ 17.73 | 62.27 | ↑ 15.86 |
| **PAPO**$_D$-7B (w/ Double Ent) | **65.83** | ↑ 15.61 | **59.82** | ↑ 19.09 | **63.16** | ↑ 17.54 |

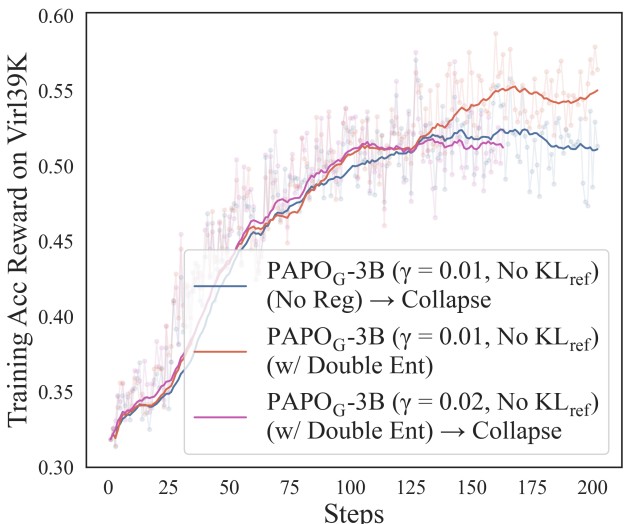

Figure 10: **Impact of KL**$_{\text{prcp}}$ **weighting ($\gamma$) under settings without reference KL.** Double Entropy Loss is indispensable for stabilizing training in this setting. Due to inherently weaker regularization, $\gamma$ should be set to a smaller value. When set higher (e.g., 0.02), model collapse still occurs, even with Double Entropy Loss.

## I    ADDITIONAL RESULTS ON ABLATION STUDIES

We provide additional results for the ablation studies discussed in § 5.2. Figure 10 presents the impact of varying the KL$_{\text{prcp}}$ weighting on PAPO$_G$ under settings with the reference KL removed. Due to inherently weaker regularization in such settings, a higher $\gamma$ (e.g., 0.02) can lead to irreversible collapse, even with Double Entropy Loss applied. Empirically, a good default $\gamma$ in this setting, including PAPO$_D$, is 0.01. Figure 10 also highlights the importance of Double Entropy Loss in enhancing training stability, even when $\gamma$ is low.

In Table 11, we further investigate the impact of the Double Entropy loss coeficients ($\eta$). Here, a smaller coefficient means less regularization strength. At 0.01, we observe model collapse in the later stages of training, whereas at 0.03 and 0.05 the entropy loss successfully prevents hacking behavior. Empirically, 0.05 performs the best across 9 evaluation benchmarks.

## J    ADDITIONAL RESULTS ON KL$_{\text{PRCP}}$ HACKING ANALYSIS

We provide additional results for the deep dive analysis on the KL$_{\text{prcp}}$ hacking problem as discussed in § 5.2. Figure 11 presents the early signs of model collapsing due to KL$_{\text{prcp}}$ Hacking. Figure 12 presents the key factors influencing the likelihood of model collapse. Figure 5 presents a comparison of the effectiveness of different regularization strategies.

Table 11: **Impact of Double Entropy Loss coefficient.** "DE" is short for Double Entropy Loss, "@ k" indicates the coefficient value for $\eta_1$ and $\eta_2$, as introduced in §3.2.

| Method | General Multimodal Reasoning | | | | | Vision-Dependent Multimodal Reasoning | | | | Overall |
|---|---|---|---|---|---|---|---|---|---|---|
| | Geo3k | MathVista | We-Math | MMKI2 | MathVerse | LogicVista | Counting | MMMU-Pro | MathVerse$_V$ | AVG |
| GRPO | 40.18 | 65.48 | **68.12** | 72.26 | 66.51 | 45.62 | 73.94 | 35.17 | 61.71 | 58.78 |
| PAPO$_G$ + DE | | | | | | | | | | |
| @ 0.01 (collapsed) | 39.95 | 62.03 | 65.19 | 69.71 | 62.72 | 44.30 | **92.50** | **38.01** | 59.77 | 59.35 |
| @ 0.03 | 40.00 | 68.09 | 67.59 | 72.43 | 68.28 | **46.46** | 89.94 | 35.85 | 61.17 | 61.09 |
| @ 0.05 | **40.25** | **69.53** | 66.79 | **72.52** | **68.43** | 46.07 | 89.81 | 36.63 | **64.97** | **61.66** |

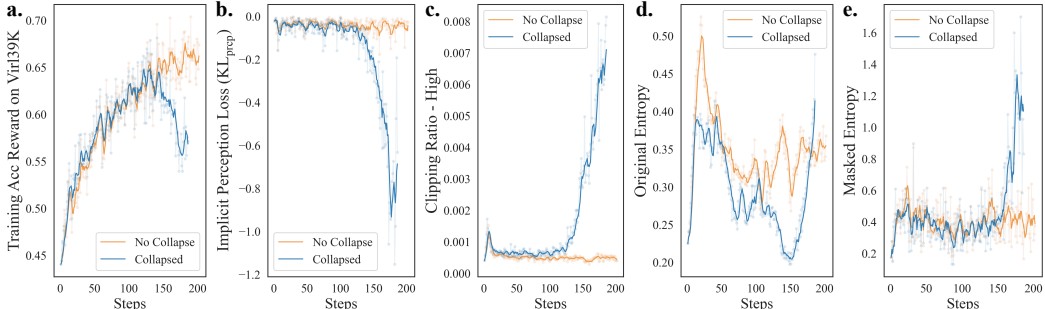

Figure 11: **Early signs of model collapsing due to KL$_{prcp}$ Hacking.** The "No Collapse" and "Collapsed" models refer to PAPO$_G$-7B ($\gamma = 0.01$) and PAPO$_G$-7B ($\gamma = 0.02$ without double entropy regularization), respectively. When collapsing occurs, we notice **(a-b)** the Implicit Perception Loss drops drastically, accompanied by a collapsing training reward, **(c)** the clipping ratio-high continuously increases, which indicates the proportion of tokens undergoing policy gradient updates beyond the clipped threshold and **(d-e)** the entropy loss increases in both the masked policy $\pi_\theta^{mask}$ and the original policy $\pi_\theta$.

## K ADDITIONAL RESULTS ON OCR TASKS

To further investigate the model's perception-centric ability, we include additional evaluation on OCR-bench-v2 (Fu et al., 2024). Table 12 shows that PAPO also brings benefits on such information-dense tasks which require fine-grained perception on the visual inputs.

## L ROBUSTNESS IN LOW VISION-DEPENDENT TASKS

Table 13 presents the results on MMLU-Pro with dummy visual inputs, which guarantees that the image is irrelevant to the query. We find that PAPO achieves competitive or better performance in this extreme setting, indicating PAPO 's robustness to noisy or uninformative visual inputs.

## M EXPLORATION ON COLLAPSING BEHAVIORS

In addition to a notable decline in performance (Table 4), collapsed models also generate responses containing entirely unrelated tokens. To explore the extent of this abnormal generation behavior, we extend our analysis to token relevance in model outputs after collapsing occurs. Qualitative examples and quantitative results can be found in Figure 13.

**Experimental Setup.** We evaluated both a collapsed 7B PAPO model ($\gamma = 0.02$, without regularization) and a non-collapsed 7B GRPO baseline on Geo3K (Lu et al., 2021). To assess the coherence and relevance of generated responses, we employed GPT-4.1-mini (OpenAI, 2025) as a judge to evaluate how well each model's response relates to and addresses the input question on a scale from 0 to 10. Our evaluation prompt below specifically instructs the judge to focus on whether the response *attempts to solve the given problem* rather than *correctness*, with scoring guidelines ranging from 0 (completely unrelated/gibberish) to 10 (perfectly related reasoning, even if the final answer is incorrect). The complete evaluation prompt is provided in Figure 14.

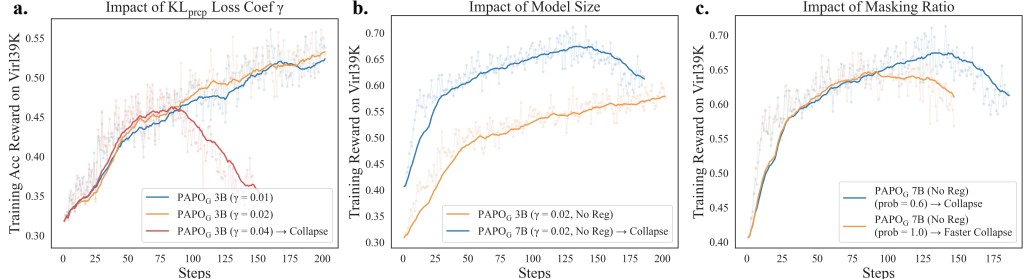

Figure 12: **Influential factors towards KL$_{\text{prcp}}$ Hacking.** We identify three main factors: **(a)** KL$_{\text{prcp}}$ weighting (higher values lead to a greater likelihood of collapse); **(b)** size (the larger the model, the more likely it is to collapse); **(c)** an extreme masking ratio (e.g., 1.0) results in a faster collapse.

Table 12: **Results on OCR-bench-v2.** The base model is Qwen2.5-VL.

| Size | Method | OCR-bench-v2 (%) |
|---|---|---|
| 3B | Base | 20.14 |
| 3B | GRPO | 26.32 |
| 3B | **PAPO$_G$** | **27.22** |
| 7B | Base | 27.75 |
| 7B | GRPO | 31.17 |
| 7B | **PAPO$_G$** | **32.11** |

Table 13: **Results on MMLU-Pro with dummy visual inputs.** The base model is Qwen2.5-VL.

| Size | Method | MMLU-pro Dummy Visual (%) |
|---|---|---|
| 3B | Base | 23.23 |
| 3B | GRPO | 39.34 |
| 3B | **PAPO$_G$** | **40.74** |
| 7B | Base | 31.57 |
| 7B | GRPO | **50.76** |
| 7B | **PAPO$_G$** | 50.55 |

**Quantitative Analysis.** Our relatedness evaluation revealed that the collapsed model demonstrated significantly degraded coherence, with an average relatedness score approximately 18% lower than the baseline model. This substantial drop in relevance scores reflects the model's tendency to generate tokens that bear little semantic relationship to the input context or fail to attempt solving the given problem. Additionally, we measured the variance in the KL$_{\text{prcp}}$ loss across response tokens by computing per-token KL divergence between responses generated from original images versus randomly patch-blackened versions of the same images. The collapsed model showed approximately 8.4 times higher variance in KL divergence compared to the baseline, indicating that the model has learned to exploit the KL$_{\text{prcp}}$ by generating highly unpredictable token sequences.

**Qualitative Observations.** Based on the grading of GPT-4.1-mini and our manual inspection, we find that collapsed models frequently generate responses that may begin with some problem-relevant content but contain substantial portions of irrelevant text, numbers, or apparent meaningless contents. An example can be found in Figure 13.

# N  ADDITIONAL ANALYSIS ON REDUCED PERCEPTION ERROR

In Figure 1, we show that PAPO significantly reduces perception-related errors. We further analyze how these instances behave: do they all become correct, or do they still struggle with other types of errors? To investigate this, we sample 60 instances that originally resulted in perception errors under the GRPO baseline and examine their outcomes under PAPO. As shown in Figure 15 41% of the original perception errors are corrected, 24% remain perception errors, and 25% shift to other error categories. This indicates that eliminating perception errors alone may not be sufficient for some challenging instances. However, overall, PAPO achieves a substantial 40% complete error removal, demonstrating strong effectiveness in improving the overall performance.

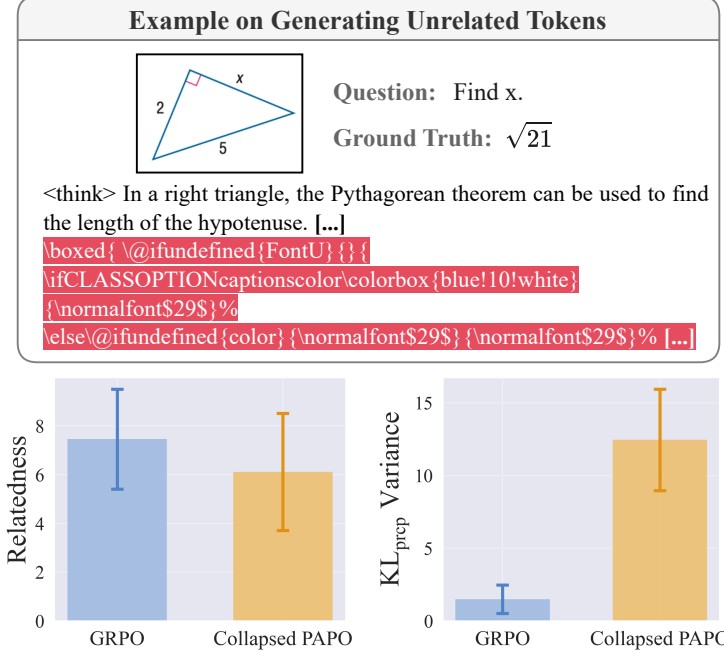

Figure 13: **Collapsing behavior.** A distinctive generation pattern in collapsed models is the production of irrelevant tokens. We verify this quantitatively by prompting GPT-4.1-mini OpenAI (2025) to provide relatedness scores of the responses from 0 to 10 for GRPO and collapsed $\text{PAPO}_G$-7B ($\gamma = 0.02$, no regularization) model. We further compare the variance of $\text{KL}_{\text{prcp}}$ over the response tokens. As illustrated, the collapsed PAPO model exhibits significantly lower relatedness scores and higher variance across tokens in $\text{KL}_{\text{prcp}}$. See §5.3 for a detailed discussion.

Table 14: Analysis of computational overhead. We report the average time per training step (in seconds) and the time spent on the additional forward pass in PAPO. The experiments are conducted using 2 and 4 NVIDIA H100 GPUs for the 3B and 7B models, respectively. While we observe a moderate increase in training time per step (67.2 seconds for 3B and 108.6 seconds for 7B) we leave further optimization of training efficiency to future work.

| Model | Method | Per Step (s) | Additional Forward Pass (s) |
|---|---|---|---|
| 3B | GRPO | 360.9 | - |
|  | $\text{PAPO}_G$ | 428.1 | 48.8 |
| 7B | GRPO | 258.5 | - |
|  | $\text{PAPO}_G$ | 367.1 | 49.7 |

## O  COMPUTATIONAL OVERHEAD ANALYSIS

The main computational overhead stems from the additional forward pass on the rollout sequences with a corrupted visual input. In Table 14, we report the averaged wall-clock time for each training step and the additional forward pass comparing GRPO and $\text{PAPO}_G$. 3B and 7B experiments are conducted on 2 and 4 Nvidia H100 GPUs respectively.

## P  ADDITIONAL ANALYSIS ON RELATED WORK

In Table 15, we additionally offer a comprehensive categorization and comparison of recent literature within the broader area of advancing multimodal reasoning. We highlight our novelty as the first work to focus on improving the optimization objectives in multimodal RL.

---

**Prompt for Scoring Relatedness**

You are an expert evaluator for mathematical visual reasoning tasks. Your job is to evaluate how well a model's reasoning and answer relates to a given question.

You will be provided with:
1. An image containing a problem
2. The question with specific response format instructions
3. The ground truth answer
4. The model's generated answer

Your task is to rate the model's answer on a scale of 0-10 based on how well it relates to and addresses the question:

**Scoring Guidelines:**
In general, the higher your score is, the more strongly the model's answer pertains to and addresses the question.
- **0-1**: Completely unrelated (talks about other irrelevant things or topics)
- **2-3**: Minimally related (mentions some relevant terms but most contents are not attempting to solve the problem)
- **4-5**: Somewhat related (attempts to address the problem but with major unrelated contents)
- **6-7**: Moderately related (addresses the problem with some correct reasoning but significant unrelated sentences or words)
- **8-9**: Highly related (good attempt at solving the problem with mostly pertaining reasoning, minor unrelated sentences or words)
- **10**: Perfectly related (correctly addresses the problem with totally related reasoning, even if final answer differs from ground truth)
Please differentiate between scores of each level (e.g., 2 is more unrelated than 3)

**Important Notes:**
- Focus on whether the answer ATTEMPTS to solve the given problem, not just correctness
- A mathematically incorrect answer can still score high if it shows proper understanding and reasoning
- Consider the formatting requirements (like using <think> tags or \boxed{{}}) as part of following instructions
- The ground truth is provided for context, but your score should be based on relevance and reasoning quality, not exact matching

**Question Text:** {formatted_question}

**Ground Truth Answer:** {ground_truth}

**Model's Generated Answer:** {generated_answer}

Based on the image, question, and model's response, provide a score from 0-10 indicating how well the model's answer relates to and addresses the question.

Respond with ONLY the numerical score (0-10), nothing else.

---

Figure 14: Prompt to GPT-4.1-mini for scoring the relatedness between the model-generated response and the input query.

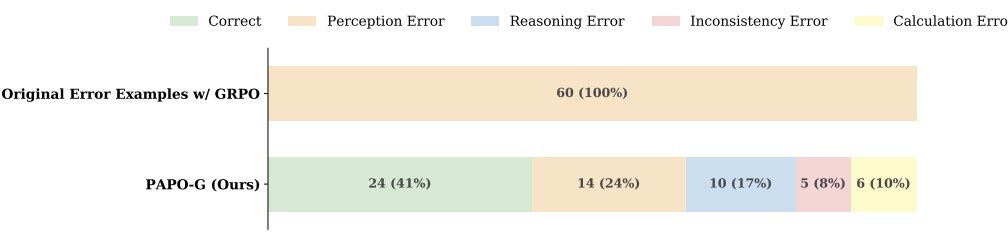

Figure 15: Detailed analysis of 60 GRPO perception-error cases and their outcomes under PAPO.

Table 15: **Comprehensive categorization and comparison of different approaches.** We highlight our novelty as the first work to focus on improving the optimization objectives in multimodal RL.

| Method | Improvement Perspective | Highlight |
|---|---|---|
| Think or Not? (Wang et al., 2025c) | Training Framework | Introduces a Cold-Start SFT stage with "thought dropout," allowing the model to skip unnecessary thinking on simple tasks to improve efficiency. |
| Semi-off-Policy (Shen et al., 2025b), Vision-SR1 (Li et al., 2025c) | Training Framework | Separates perception and reasoning into two stages/models: generates captions first, then performs reasoning on them, and propagates rewards back to improve the multimodal model. |
| Open Vision Reasoner (Wei et al., 2025), R1-V (Chen et al., 2025), LMM-R1 (Peng et al., 2025) | Data | Synthesizes reasoning trajectories by leveraging strong text-only reasoning models such as DeepSeek-R1. |
| Reason-RFT (Tan et al., 2025), MM-Eureka (Meng et al., 2025b), VL-rethinker (Wang et al., 2025b) | Data | Constructs a diverse multimodal reasoning dataset covering counting, structural perception, and spatial transformation. |
| NoisyRollout (Liu et al., 2025a), R1-ShareVL (Yao et al., 2025) | Rollout | Introduces semantic-preserving data augmentation during rollout, such as Gaussian noise, to diversify exploration and improve robustness. |
| VL-rethinker (Wang et al., 2025b), Skywork R1V2 (Wang et al., 2025d) | Rollout | Introduces Selective Sample Replay to mitigate the issue of vanishing advantages. |
| Visual-RFT (Liu et al., 2025c), Vision-R1 (Huang et al., 2025), V-Triune (Ma et al., 2025), Perception-R1 (Xiao et al., 2025b), VisionReasoner (Liu et al., 2025b) | Reward | Introduces grounding-related rewards such as IoU and Dynamic IoU to enhance detection and localization performance. |
| Visionary-R1 (Xia et al., 2025) | Reward | Introduces model-based rewards computed directly on captions generated by the policy model. |
| GRIT (Fan et al., 2025), TreeVGR (Wang et al., 2025a), DeepEyes (Zheng et al., 2025), Mini-o3 (Lai et al., 2025) | Reward | Introduces multi-turn rewards using "thinking-with-images" or "thinking-with-bounding-boxes" paradigms. |
| **PAPO (ours)** | Optimization Objective | Distinct from previous work, PAPO explores improvements from the core optimization objective tailored to the multimodal domain, and shows compatibility with advances from other perspectives. |

