# OpenReview forum: "Perception-Aware Policy Optimization for Multimodal Reasoning"
_ICLR.cc/2026/Conference — ICLR 2026 Poster_

### Official Review · Reviewer_BkD7 · 2025-10-16

**Soundness:** 4
**Presentation:** 4
**Contribution:** 3
**Rating:** 6
**Confidence:** 5

**Summary:**

This paper identifies a critical bottleneck in large multimodal models (LMMs): most failures in complex reasoning tasks arise from visual perception errors rather than logical flaws. To address this, the authors propose Perception-Aware Policy Optimization (PAPO), a novel algorithm that extends Reinforcement Learning with Verifiable Rewards (RLVR) frameworks such as GRPO and DAPO.
PAPO introduces two key components:
Implicit Perception Loss, which encourages visually grounded responses by maximizing the KL divergence between output distributions conditioned on original versus corrupted visual inputs.
Double Entropy Loss, which regularizes the objective and improves training stability.
Experiments on eight multimodal reasoning benchmarks show a 4.4%–17.5% overall improvement and a 30.5% reduction in perception errors.

**Strengths:**

- Clear and Important Motivation: The work is well-grounded in a detailed error analysis showing that perception errors account for the majority of LMM failures. This provides strong motivation for the proposed approach.
- Novel and Efficient Design: PAPO’s use of an implicit “reverse KL” objective to enforce visual grounding is elegant and requires no extra annotations, reward models, or teacher models, making it an efficient drop-in replacement for existing RLVR methods.
- Comprehensive Experiments: The paper evaluates multiple benchmarks and model sizes, includes ablations on masking strategies and loss weighting, and provides in-depth analysis of failure modes and stability mechanisms.

**Weaknesses:**

- Flawed Core Assumption: The hypothesis that model confidence should always decrease when visual information is removed is not universally valid. Robust models should ignore irrelevant visual cues, meaning PAPO may incorrectly penalize robustness in cluttered or partially relevant visual scenes (Although their mask strategies seem good).
- High Computational Overhead: PAPO increases per-step training time by over 40% for the 7B model, while the accuracy gains are often modest, making the cost–benefit ratio questionable.
- Metric-Method Collusion Risk: The primary evaluation metric (accuracy@8) can be artificially inflated by low-entropy policies. PAPO’s entropy regularizer may bias results toward deterministic outputs rather than genuine reasoning improvements.
- Counterintuitive Ablation Results: The finding that random masking outperforms semantic masking suggests the method may be learning brittle correlations rather than true visual grounding, can authors make more comparisons on this problem.

**Questions:**

Please finish the concerns in the weakness

---

> ### Author Response · Authors · 2025-11-21
>
> **We thank reviewer BkD7 for the detailed and insightful comments. We appreciate the reviewer’s acknowledgement of our strengths and for contributing to a positive and professional discussion. We address the remaining questions and suggestions below. We will update the manuscript reflecting these changes before Dec 2.**
>
> # W1: Further analysis of PAPO’s effectiveness in visually irrelevant scenarios
> We fully acknowledge the reviewer’s concern. As also discussed in Section 7, the current method applies the implicit perception loss to all instances and all tokens. Although this minimalist design works empirically well, we agree that there is substantial room for improvement through a more sophisticated gating or token-selection mechanism to determine which tokens or instances should receive the loss.
>
> To further examine whether the current PAPO model remains effective in strictly vision-independent scenarios, we **conducted an additional evaluation experiment**. Specifically, we **take a widely used text-only benchmark, MMLU-pro [1], and insert dummy visual inputs into the context** (images containing pure random noise). This setup ensures that the **visual input is entirely irrelevant to the question.** If the model were to attend to the noisy visual tokens indiscriminately, we would expect a degradation in performance.
>
> The results comparing PAPO and GRPO baselines are shown below:
>
> | Model (Qwen2.5-VL) | MMLU-Pro + Dummy Visual: Acc (%) |
> |-|-|
> | 3B Base | 23.23 |
> | 3B GRPO | 39.34 |
> | 3B PAPO-G | 40.74 |
> |
> | 7B Base | 31.57 |
> | 7B GRPO | 50.76 |
> | 7B PAPO-G | 50.55 |
>
> We find that **PAPO still achieves competitive or stronger performance under this extreme setting**, indicating good robustness in low vision-dependency scenarios.
>
> ### Reference for W1:
> - [1] Wang, Yubo, et al. "Mmlu-pro: A more robust and challenging multi-task language understanding benchmark." Advances in Neural Information Processing Systems 37 (2024): 95266-95290.
>
>
> # W2: Computation overhead
> We agree that there is a noticeable computational overhead when adding PAPO. However, PAPO also achieves faster convergence, as shown in Figure 4. For example, on the 3B model, PAPO-G reaches the same level of training reward at around 150 steps, whereas GRPO requires about 200 steps. In addition, the computational overhead occurs only during training and does not affect inference. Nevertheless, we acknowledge that there is still substantial room for improving training efficiency, and consider this an important direction for future work.

---

> > ### Author Response · Authors · 2025-11-21
> >
> > # W3: Metric-method collusion risk
> > The reviewer raises an interesting question regarding the choice of evaluation metric. We **follow mainstream prior work, such as DeepSeek-R1 [1], DAPO [2], and Skywork-R1V2 [3], in using Avg@K (i.e., Pass@1 over K independent generations with a non-zero temperature) as our evaluation metric.** We believe this remains the most widely accepted metric in recent reasoning-focused literature. Another alternative adopted by some recent work [4,5], such as NoisyRollout, is Greedy@1 (generating a single response using a near-zero temperature). However, that would make the inference even more deterministic, so we did not adopt it.
> >
> > The reviewer also raises an important discussion about the relationship between entropy and reasoning ability. One commonly accepted view is that the RL stage essentially **“trades” entropy for reasoning performance**, a perspective extensively studied in [6,7]. From this viewpoint, we would not say that adding an entropy-based regularization directly prevents “genuine reasoning improvements.”
> >
> > From another perspective, recent work has explored the **benefits of maintaining higher entropy during training**. For example, [8] proposes using Pass@K instead of Pass@1 as the reward signal, and [9] specifically prioritizes updating high-entropy tokens while ignoring low-entropy ones. In our setting, we find that the **PAPO objective tends to increase the entropy** of the generations; without proper regularization, this can lead to model collapse. Therefore, we **introduce an entropy loss to counteract this effect.**
> >
> > Overall, we believe this remains an active and open research question that is worth deeper investigation.
> >
> > ### Reference for W3:
> > - [1] Guo, Daya, et al. "Deepseek-r1: Incentivizing reasoning capability in llms via reinforcement learning." arXiv preprint arXiv:2501.12948 (2025).
> > - [2] Yu, Qiying, et al. "Dapo: An open-source llm reinforcement learning system at scale." arXiv preprint arXiv:2503.14476 (2025).
> > - [3] Wang, Peiyu, et al. "Skywork r1v2: Multimodal hybrid reinforcement learning for reasoning." arXiv preprint arXiv:2504.16656 (2025).
> > - [4] Liu, Xiangyan, et al. "Noisyrollout: Reinforcing visual reasoning with data augmentation." arXiv preprint arXiv:2504.13055 (2025).
> > - [5] Visionary-r1: Mitigating shortcuts in visual reasoning with reinforcement learning.
> > - [6] Cui, Ganqu, et al. "The entropy mechanism of reinforcement learning for reasoning language models." arXiv preprint arXiv:2505.22617 (2025).
> > - [7] Prabhudesai, Mihir, et al. "Maximizing Confidence Alone Improves Reasoning." arXiv preprint arXiv:2505.22660 (2025).
> > - [8] Chen, Zhipeng, et al. "Pass@ k training for adaptively balancing exploration and exploitation of large reasoning models." arXiv preprint arXiv:2508.10751 (2025).
> > - [9] Wang, Shenzhi, et al. "Beyond the 80/20 rule: High-entropy minority tokens drive effective reinforcement learning for llm reasoning." arXiv preprint arXiv:2506.01939 (2025).

---

> ### Author Response · Authors · 2025-11-21
>
> # W4: Deeper analysis on the masking strategy
> As suggested by the reviewer, we conducted further analysis trying to understand the counterintuitive empirical observation that the random-masking performs better than semantic-masking.
>
> We believe random masking performs well because it naturally strikes a balance between masking out enough informative content while avoiding the removal of an entire “block” of salient information.
> A similar phenomenon has been observed in prior work on **masked autoencoders [1] and masked visual modeling [2,3]**, where random masking is found to be competitive with, or even superior to, block-wise and attention-based masking strategies. However, understanding the underlying reason beyond these empirical observations remains non-trivial.
>
> During the rebuttal, we made **two initial attempts to further understand this question**:
> (1) analyzing the attention patterns of PAPO models trained with random masking versus semantic masking, and
> (2) examining the error distribution in cases where the random-masking model succeeds but the semantic-masking model fails.
>
> ## W4.1: Findings from attention-pattern analysis
> Our goal is to investigate whether the masking strategy leads to different attention behaviors after training. We extract the average attention weight from response tokens to input image tokens, and then compute the normalized entropy of the attention distribution across the image tokens. The normalized entropy is defined as Ent(P) / Ent(P_uni), where P_uni is the uniform distribution over image patches; this normalization accounts for different numbers of image tokens. A higher entropy indicates a more scattered attention distribution, whereas a lower entropy indicates a more centralized one. We conduct this experiment on two datasets, Geo3K and Counting, sampling 100 instances per dataset.
>
>
> | Model (Qwen2.5-VL) | Mean - Geo3K  | Median - Geo3K  | Mean - Counting  | Median - Counting  |
> |-|-|-|-|-|
> | 3B PAPO-G w/ Random-Masking  | 0.9520 | 0.9522 | 0.9351 | 0.9362
> | 3B PAPO-G w/ Semantic-Masking  | 0.9522 | 0.9517 | 0.9331 | 0.9342
> | 7B PAPO-G w/ Random-Masking  | 0.9457 | 0.9472 | 0.9525 | 0.9530
> | 7B PAPO-G w/ Semantic-Masking  | 0.9401 | 0.9403 | 0.9513 | 0.9512
>
> As shown in the table above, we find that the overall attention patterns **do not differ significantly** between the two masking strategies. However, the random-masking model **generally exhibits slightly higher entropy**, indicating a more scattered attention distribution. This observation aligns with our hypothesis that semantic masking encourages the model to focus more heavily on a centralized salient region.
>
> ## W4.2: Findings from error analysis
> We also investigate **what types of errors cause the semantic-masking model to underperform relative to the random-masking model**. We sample 30 instances per dataset from Geo3K, Counting, and MathVerse where the random-masking model is correct but the semantic-masking model fails, and manually categorize the error types.
>
> | Semantic-masking Failure Modes | Perception Error  | Reasoning Error  | Inconsistency Error  | Calculation Error |
> |-|-|-|-|-|
> Geo3K | 73.33% | 6.67% | 3.33% | 16.67%
> Counting | 93.33% | 0.00% | 6.67% | 0.00%
> MathVerse | 76.67% | 6.67% | 6.67% | 10.00%
>
> We find that the **predominant error source is still perception-related errors**, suggesting that the primary advantage of random-masking over semantic-masking lies in improved perception ability.
>
> ### Reference for W4:
> - [1] He, Kaiming, et al. "Masked autoencoders are scalable vision learners." Proceedings of the IEEE/CVF conference on computer vision and pattern recognition. 2022.
> - [2] Xie, Zhenda, et al. "Simmim: A simple framework for masked image modeling." Proceedings of the IEEE/CVF conference on computer vision and pattern recognition. 2022.
> - [3] Fu, Tsu-Jui, et al. "An empirical study of end-to-end video-language transformers with masked visual modeling." Proceedings of the IEEE/CVF Conference on Computer Vision and Pattern Recognition. 2023.

---

> ### Comment · Reviewer_BkD7 · 2025-11-25
> **Thank you for your response**
>
> Thanks for the great rebuttal and work from authors, all my concerns have been solved. I think this paper should be accepted due to the extensive and comprehensive experiments. All the motivation and insight are good enough.

---

> > ### Author Response · Authors · 2025-11-25
> >
> > We thank reviewer BkD7 again for the professional comments that further strengthened our submission!

---

### Official Review · Reviewer_zdmV · 2025-10-27

**Soundness:** 4
**Presentation:** 3
**Contribution:** 3
**Rating:** 8
**Confidence:** 4

**Summary:**

The paper proposes Perception-Aware Policy Optimization (PAPO), a new reinforcement learning framework for large multimodal models that integrates perceptual awareness directly into policy optimization. Unlike prior works, PAPO jointly learns both perception and reasoing within the core RL objective. The key idea is an implicit perception loss which encourages visually grounded reasoning without external supervision. A double entropy regularizer is introduced to stabilize training. PAPO serves as a plug-in-play replacement for GRPO/DAPO and achieves 4.4-17.5% average improvement across eight multimodal reasoning benchmarks, with up to 19.1% gain on vision-intensive tasks and 30.5% fewer perception-related errors. The method is simple, annotation-free and computationally efficient, showing faster convergence and more robust multimodal understanding

**Strengths:**

1. The paper introduces a novel idea of integrating perception awareness directly into the policy optimization objective, moving beyond traditional data- or reward-level modifications in multimodal RL.

2. The methodology is well-motivated, theoretically sound, and empirically validated. The authors provide thorough experiments across eight multimodal reasoning benchmarks, showing consistent and interpretable gains over strong baselines such as GRPO and DAPO. The analysis includes detailed ablations and error breakdowns, supporting the claimed contributions.

3. The paper is clearly written and logically organized. The motivation, derivation of the objective, and implementation details are easy to follow. Figures and examples effectively illustrate both intuition and quantitative results.

**Weaknesses:**

1. The experiments are conducted only on the Qwen2.5‑VL model (3B/7B) and do not include other pretrained models (e.g., InternVL).

2. The paper lacks a discussion of scalability. Specifically, whether the proposed method remains effective (and efficient) when applied to much larger models (e.g., ~30B-70B parameters).

3. The paper could porvide a deeper discussion of how the proposed method differs (in objective, reward design, computational cost, etc.) from those works would improve the clarity of novelty:

  [1] Wang, Jiaqi; Lin, Kevin Qinghong; Cheng, James; Shou, Mike Zheng. Think or Not? Selective Reasoning via Reinforcement Learning for Vision-Language Models.

  [2] Shen, Junhao; Zhao, Haiteng; Gu, Yuzhe; Gao, Songyang; Liu, Kuikun; Huang, Haian; Gao, Jianfei; Lin, Dahua; Zhang, Wenwei; Chen, Kai. Semi-off-Policy Reinforcement Learning for Vision-Language Slow-Thinking Reasoning.

  [3] Wei, Yana; Zhao, Liang; Sun, Jianjian; Lin, Kangheng; Yin, Jisheng; Hu, Jingcheng; Zhang, Yingmin; Weng, Zejia; Wang, Jia; Han, Chunrui; Peng, Yuang; Han, Qi; Ge, Zheng; Zhang, Xiangyu; Jiang, Daxin; Patel, Vishal M. Open Vision Reasoner: Transferring Linguistic Cognitive Behavior for Visual Reasoning.

  [4] Liu, Ziyu; Sun, Zeyi; Zang, Yuhang; Dong, Xiaoyi; Cao, Yuhang; Duan, Haodong; Lin, Dahua; Wang, Jiaqi. Visual Reinforcement Fine-Tuning.

  [5] Tan, Huajie; Ji, Yuheng; Hao, Xiaoshuai; Lin, Minglan; Wang, Pengwei; Wang, Zhongyuan; Zhang, Shanghang. Reason-RFT: Reinforcement Fine-Tuning for Visual Reasoning.

**Questions:**

Could the authors clarify why maximizing the reverse KL divergence between rollouts with original and masked visual inputs improves perception–reasoning coupling? A brief theoretical or deeper empirical explanation would help justify this design choice.

Apart from this question, I have no further concerns.

---

> ### Author Response · Authors · 2025-11-21
>
> **We thank reviewer zdmV for the detailed and insightful comments. We appreciate the reviewer’s acknowledgement of our strengths and for contributing to a positive and professional discussion. We address the remaining questions and suggestions below. We will update the manuscript reflecting these changes before Dec 2.**
>
> # W1: Experiments conducted only on Qwen2.5-VL models
>
> We understand the reviewer’s concern, which also echoed our discussion in the limitations section. We chose to focus on the Qwen2.5-VL family because it is currently the most widely used and infrastructure-supported model series, and it has been adopted as the only base model in the majority of recent multimodal RLVR literature [1,2,3,4,5]. Therefore, we believe that conclusions drawn from experiments on Qwen2.5-VL remain robust and well-supported by the community.
>
> Nevertheless, we agree that including results on broader model series would further enhance the evidence of the effectiveness. Thus we try to apply PAPO to Qwen3-VL and InternVL3 models:
>
> **New results on Qwen3-VL**: We conduct experiments on the newly released Qwen3-VL-2B-thinking model. Architecturally, Qwen3-VL differs from Qwen2.5-VL through the use of the DeepStack strategy in its vision encoder and a new Interleaved-MRoPE positional embedding.
>
>
> | Model (Qwen3-VL-2B-thinking) | Geo3K | MathVista | We-Math | MMK12 | MathVerse | LogicVista | Counting | MMMU-Pro | MathVerse-V | AVG
> |-|-|-|-|-|-|-|-|-|-|-|
> | GRPO | 39.29 | 53.58 | 57.12 | 47.71 | 47.98 | 29.84 | 80.13 | 20.51 | 45.41 | 46.84
> | PAPO-G | 41.08↑ | 56.08↑ | 59.17↑ | 48.57↑ | 51.89↑ | 32.83↑ | 80.63↑ | 23.42↑ | 50.05↑ | 49.30↑
>
> As shown in the Table, we see consistent improvements against the GRPO baseline, demonstrating a strong generalization capability to different model architectures.
>
> **InternVL3 attempt**: We also attempted to incorporate the InternVL3 series into our codebase, which is not yet supported by the EasyR1 framework. However, we encountered kernel-related errors that we are unlikely to resolve within the rebuttal period. We will continue reaching out to the EasyR1 and vLLM maintainers to further investigate the issue.
>
> ### Reference for W1:
> - [1] Liu, Xiangyan, et al. "Noisyrollout: Reinforcing visual reasoning with data augmentation." arXiv preprint arXiv:2504.13055 (2025).
> - [2] Wang, Haozhe, et al. "Vl-rethinker: Incentivizing self-reflection of vision-language models with reinforcement learning." arXiv preprint arXiv:2504.08837 (2025).
> - [3] Yao, Huanjin, et al. "R1-ShareVL: Incentivizing Reasoning Capability of Multimodal Large Language Models via Share-GRPO." arXiv preprint arXiv:2505.16673 (2025).
> - [4] Yang, Yi, et al. "R1-onevision: Advancing generalized multimodal reasoning through cross-modal formalization." arXiv preprint arXiv:2503.10615 (2025).
> - [5] Xia, Jiaer, et al. "Visionary-r1: Mitigating shortcuts in visual reasoning with reinforcement learning." arXiv preprint arXiv:2505.14677 (2025).

---

> ### Author Response · Authors · 2025-11-21
>
> # W2: Lack of scaling to larger model size (e.g., 30B~70B)
> We acknowledge the reviewer’s concern and have also discussed this in our limitation section. Unfortunately, due to limited computing resources in academia, the largest model size we are able to tune is 7B. We agree that validating the idea on larger-scale models is an important direction for future work.
>
> # W3: Deeper discussion on comparison with recent related work
> We thank the reviewer for providing pointers to additional recent literature. We offer a **comprehensive categorization and comparison of different approaches** within the broader area of advancing multimodal reasoning.
>
> We highlight our novelty as **the first work to focus on improving the optimization objectives in multimodal RL**.
>
> Method | Improvement Perspective | Highlight
> |-|-|-|
> Think or Not?[1]                                                   | Training Framework | Introduces a Cold-Start SFT stage with “thought dropout,” allowing the model to skip unnecessary thinking on simple tasks to improve efficiency.
> Semi-off-Policy[2]; Vision-SR1[3]                                     | Training Framework | Separates perception and reasoning into two stages/models: generates captions first, then performs reasoning on them, and propagates rewards back to improve the multimodal model.
> Open Vision Reasoner[4]; R1-V[5]; LMM-R1[6]                              | Data | Synthesizes reasoning trajectories by leveraging strong text-only reasoning models such as DeepSeek-R1.
> Reason-RFT[7]; MM-Eureka[8]; VL-rethinker[9]                             | Data | Constructs a diverse multimodal reasoning dataset covering counting, structural perception, and spatial transformation.
> NoisyRollout[10]; R1-ShareVL[11]                                        | Rollout | Introduces semantic-preserving data augmentation during rollout, such as Gaussian noise, to diversify exploration and improve robustness.
> VL-rethinker[9], Skywork R1V2[12]                                      | Rollout | Introduces Selective Sample Replay mechanism to mitigate the issue of vanishing advantages.
> Visual-RFT[13]; Vision-R1[14]; V-Triune[15]; Perception-R1[16]; VisionReasoner[17]  | Reward | Introduces grounding-related rewards such as IoU and dynamic IoU to enhance detection and localization performance.
> Visionary-R1[18]                                                    | Reward | Introduces model-based rewards computed directly on captions generated by the policy model.
> GRIT[19]; TreeVGR[20]; DeepEyes[21]; Mini-o3[22]                                | Reward | Introduces multi-turn rewards using “thinking-with-images” or “thinking-with-bounding-boxes” paradigms.
> |  |
> **PAPO (ours)**                                                            | Optimization Objective | Distinct from previous work, PAPO explores improvements on the core optimization objective tailored to the multimodal domain, and shows PAPO can integrate with advances from other perspectives.

---

> > ### Author Response · Authors · 2025-11-21
> >
> > ### Reference for W3:
> > - [1] Wang, Jiaqi, et al. "Think or Not? Selective Reasoning via Reinforcement Learning for Vision-Language Models." arXiv preprint arXiv:2505.16854 (2025).
> > - [2] Shen, Junhao, et al. "Semi-off-Policy Reinforcement Learning for Vision-Language Slow-thinking Reasoning." arXiv preprint arXiv:2507.16814 (2025).
> > - [3] Li, Zongxia, et al. "Self-rewarding vision-language model via reasoning decomposition." arXiv preprint arXiv:2508.19652 (2025).
> > - [4] Wei, Yana, et al. "Open vision reasoner: Transferring linguistic cognitive behavior for visual reasoning." arXiv preprint arXiv:2507.05255 (2025).
> > - [5] Liang Chen, et al. R1-v: Reinforcing super generalization ability in vision-language models with less than $3. https://github.com/Deep-Agent/ (2025).
> > - [6] Peng, Yingzhe, et al. "Lmm-r1: Empowering 3b lmms with strong reasoning abilities through two-stage rule-based rl." arXiv preprint arXiv:2503.07536 (2025).
> > - [7] Tan, Huajie, et al. "Reason-rft: Reinforcement fine-tuning for visual reasoning." arXiv preprint arXiv:2503.20752 (2025).
> > - [8] Meng, Fanqing, et al. "Mm-eureka: Exploring the frontiers of multimodal reasoning with rule-based reinforcement learning." arXiv preprint arXiv:2503.07365 (2025).
> > - [9] Wang, Haozhe, et al. "Vl-rethinker: Incentivizing self-reflection of vision-language models with reinforcement learning." arXiv preprint arXiv:2504.08837 (2025).
> > - [10] Liu, Xiangyan, et al. "Noisyrollout: Reinforcing visual reasoning with data augmentation." arXiv preprint arXiv:2504.13055 (2025).
> > - [11] Yao, Huanjin, et al. "R1-ShareVL: Incentivizing Reasoning Capability of Multimodal Large Language Models via Share-GRPO." arXiv preprint arXiv:2505.16673 (2025).
> > - [12] Wang, Peiyu, et al. "Skywork r1v2: Multimodal hybrid reinforcement learning for reasoning." arXiv preprint arXiv:2504.16656 (2025).
> > - [13] Liu, Ziyu, et al. "Visual-rft: Visual reinforcement fine-tuning." arXiv preprint arXiv:2503.01785 (2025).
> > - [14] Zhan, Yufei, et al. "Vision-r1: Evolving human-free alignment in large vision-language models via vision-guided reinforcement learning." arXiv preprint arXiv:2503.18013 (2025).
> > - [15] Ma, Yan, et al. "One RL to See Them All: Visual Triple Unified Reinforcement Learning." arXiv preprint arXiv:2505.18129 (2025).
> > - [16] Yu, En, et al. "Perception-r1: Pioneering perception policy with reinforcement learning." arXiv preprint arXiv:2504.07954 (2025).
> > - [17] Liu, Yuqi, et al. "VisionReasoner: Unified Visual Perception and Reasoning via Reinforcement Learning." arXiv preprint arXiv:2505.12081 (2025).
> > - [18] Xia, Jiaer, et al. "Visionary-r1: Mitigating shortcuts in visual reasoning with reinforcement learning." arXiv preprint arXiv:2505.14677 (2025).
> > - [19] Fan, Yue, et al. "GRIT: Teaching MLLMs to Think with Images." arXiv preprint arXiv:2505.15879 (2025).
> > - [20] Wang, Haochen, et al. "Traceable evidence enhanced visual grounded reasoning: Evaluation and methodology." arXiv preprint arXiv:2507.07999 (2025).
> > - [21] Zheng, Ziwei, et al. "DeepEyes: Incentivizing" Thinking with Images" via Reinforcement Learning." arXiv preprint arXiv:2505.14362 (2025).
> > - [22] Lai, Xin, et al. "Mini-o3: Scaling up reasoning patterns and interaction turns for visual search." arXiv preprint arXiv:2509.07969 (2025).
> >
> >
> >
> >
> > # Q1: Further explanation of why the reverse KL divergence improves perception-reasoning coupling
> >
> > **High-level idea of the reverse KL divergence (KL_prcp):**
> > - Original GRPO objective: we encourage the model to generate responses that achieve high rewards.
> > - GRPO + KL_prcp objective: we encourage the model to generate responses that both depend on the image and achieve high rewards.
> >
> > **More formally, from an information theory perspective:**
> > Let the masked and unmasked visual inputs be I_m and I_u, respectively. Define Q = p(o | q, I_u), P = p(o | q, I_u, I_m), which denote the probability of generating a response o conditioned on the query q and different amounts of visual information.
> >
> > According to the definition of information gain, the proposed KL term KL(P || Q) corresponds exactly to an information gain, where Q is the prior and P is the posterior. The “additional observation” in the posterior P is precisely the masked image patches.
> >
> > Thus, by maximizing the proposed KL divergence, we effectively encourage the model to rely more on the masked visual information when generating responses. This results in a stronger coupling between perception and reasoning.

---

> > > ### Comment · Reviewer_zdmV · 2025-11-22
> > > **Thanks for Your Response - I Still Have a Few Questions**
> > >
> > > Thank you for your detailed response, especially for conducting the additional experiments and further distinguishing your approach from recent methods. This has alleviated my concerns about the generality of the method and strengthened my confidence in its novelty.
> > >
> > > However, I still have a few questions that I hope we can discuss further:
> > >
> > > 1. While the explanation of why maximizing the reverse KL divergence increases the reliance on visual input is clear to me, I am still wondering whether there is an experimental evidence showing that the model indeed pays more attention to the relevant image regions after applying PAPO (beyond simply improving accuracy on benchmarks). For example, visualization, such as attention maps or cross-model attention heatmaps, could directly show that the model internalizes stronger perception-reasoning coupling.
> > >
> > > 2. I understand the resource limitations regarding larger models. However, I would still like to ask about the scaling behavior of PAPO on the models you have already experimented with (3B & 7B). Although Table 1 presents results for both sizes, the paper does not analyze whether PAPO's performance gains scale with the size of model and data, i.e., whether the effect might strengthen as model or data scale increases. Such analysis would greatly help clarify whether PAPO benefits more from larger training data/model, even if experiments beyond 7b are not feasible.

---

> > > > ### Author Response · Authors · 2025-11-22
> > > >
> > > > We thank the reviewer for the additional suggestions and actively engage in the discussion!
> > > >
> > > > 1. We will add additional qualitative examples visualizing the attention heatmaps for the GRPO and PAPO models. These will be included in the revised manuscript, which we plan to upload within the next week.
> > > >
> > > > 2. We agree that further investigating scaling behavior is important. From the results currently available (3B and 7B), we observe that:
> > > >    - the overall improvements are consistent and of similar magnitude (4.36% vs. 4.39% relative gains)
> > > >    - the gains on vision-dependent tasks become more pronounced in the 7B model (7.96% vs. 5.60%)
> > > >
> > > >   For larger models, we anticipate diminishing returns as the base performance begins to saturate. We are also actively looking for additional compute to conduct more extensive scaling experiments.

---

> > > > > ### Comment · Reviewer_zdmV · 2025-11-23
> > > > >
> > > > > Thank you for the further clarification. I don't have any additional questions for now. I believe this work is also one of the few recent methods that design training strategies from a visual perspective. I'll keep my score as it is for the moment, and perhaps we can have further discussion after I read the revised paper.

---

> > > > > > ### Author Response · Authors · 2025-11-24
> > > > > >
> > > > > > Thank you for acknowledging the novelty of this work! We have just uploaded the revised paper and added the suggested attention visualization in Figure 7.

---

> > > > > > > ### Comment · Reviewer_zdmV · 2025-11-26
> > > > > > >
> > > > > > > Thank you very much for your response. The revised paper has further addressed my concerns, and I will raise my score.

---

> > > > > > > > ### Author Response · Authors · 2025-11-26
> > > > > > > >
> > > > > > > > We thank reviewer zdmV again for the professional comments that further strengthened our submission!

---

> ### Author Response · Authors · 2025-11-29
> **Additional information on reviewer zdmV's feedback**
>
> Due to the reverting of the review states, we provide additional discussion history that occurred before the incident. We aim to faithfully reflect the reviewer’s active and professional engagement during the discussion, which we greatly appreciate.
>
> - After we submitted the revised manuscript, the reviewer updated their review by marking the weaknesses and questions as “[Resolved]” and raising the score from 8 to 10.

---

### Official Review · Reviewer_XvCB · 2025-10-29

**Soundness:** 3
**Presentation:** 3
**Contribution:** 3
**Rating:** 6
**Confidence:** 4

**Summary:**

This paper proposes Perception-Aware Policy Optimization (PAPO), an RLVR algorithm. It addresses dominant visual perception errors in multimodal LLMs by integrating an Implicit Perception Loss (KL divergence with masked visual inputs) into GRPO, fostering visually grounded reasoning. A Double Entropy Loss ensures training stability against "loss hacking." PAPO achieves significant performance boosts on multimodal benchmarks, notably without external data or reward models.

**Strengths:**

1.The paper convincingly identifies and empirically quantifies visual perception errors as a primary source of failure in multimodal reasoning, providing a strong foundation for its contribution.
2. PAPO's core idea of using an Implicit Perception Loss to intrinsically encourage visually grounded reasoning, alongside a robust Double Entropy Loss for stability, is novel, intuitive, and empirically proven to yield substantial performance gains.
3. PAPO achieves state-of-the-art improvements without relying on additional data curation, external reward models, or stronger teacher models, making it a highly practical and accessible method for enhancing multimodal LLMs.

**Weaknesses:**

1.I am concerned about the result where random masking empirically outperforms DINOv2-based semantic-aware masking. The authors' hypothesis that semantic-aware masking obscures entire salient regions, leading to indiscriminate attention, requires more robust empirical validation. This outcome, coupled with the observation that pixel-level noise is less effective at obscuring informative semantics, prompts a critical need for deeper analysis into the optimal balance between information retention and strategic perturbation.
2.While the paper commendably demonstrates PAPO's effectiveness in reducing perception errors, particularly on high vision-dependency tasks, I am concerned about the potential implications for tasks with low visual dependency. Explicitly enforcing visual grounding via the Implicit Perception Loss in such scenarios might introduce unnecessary computational overhead or divert the model's attention away from pertinent textual information. I recommend that the authors conduct dedicated sub-dataset analyses on low-dependency tasks and/or perform a thorough bad-case analysis to understand PAPO's behavior in these contexts. Furthermore, while Figure 1 indicates a significant reduction in perception errors, it is imperative to analyze whether this reduction leads to a compensatory increase in other error categories (e.g., reasoning or calculation errors).

**Questions:**

Please refer to the Weaknesses section above for details.

---

> ### Author Response · Authors · 2025-11-21
>
> **We thank reviewer XvCB for the detailed and insightful comments. We appreciate the reviewer’s acknowledgement of our strengths and for contributing to a positive and professional discussion. We address the remaining questions and suggestions below. We will update the manuscript reflecting these changes before Dec 2.**
>
> # W1: Deeper analysis on the masking strategies
> The reviewer raises an interesting question: when performing masking for PAPO, how should we balance information retention with strategic perturbation?
>
> We believe random masking performs well because it naturally strikes a balance between masking out enough informative content while avoiding the removal of an entire “block” of salient information.
> A similar phenomenon has been observed in prior work on **masked autoencoders [1] and masked visual modeling [2,3]**, where random masking is found to be competitive with, or even superior to, block-wise and attention-based masking strategies. However, understanding the underlying reason beyond these empirical observations remains non-trivial.
>
> During the rebuttal, we made **two initial attempts to further understand this question**:
> (1) analyzing the attention patterns of PAPO models trained with random masking versus semantic masking, and
> (2) examining the error distribution in cases where the random-masking model succeeds but the semantic-masking model fails.
>
> ## W1.1: Findings from attention-pattern analysis.
> Our goal is to investigate whether the masking strategy leads to different attention behaviors after training. We extract the average attention weight from response tokens to input image tokens, and then compute the normalized entropy of the attention distribution across the image tokens. The normalized entropy is defined as Ent(P) / Ent(P_uni), where P_uni is the uniform distribution over image patches; this normalization accounts for different numbers of image tokens. A higher entropy indicates a more scattered attention distribution, whereas a lower entropy indicates a more centralized one. We conduct this experiment on two datasets, Geo3K and Counting, sampling 100 instances per dataset.
>
>
> | Model (Qwen2.5-VL) | Mean - Geo3K  | Median - Geo3K  | Mean - Counting  | Median - Counting  |
> |-|-|-|-|-|
> | 3B PAPO-G w/ Random-Masking  | 0.9520 | 0.9522 | 0.9351 | 0.9362
> | 3B PAPO-G w/ Semantic-Masking  | 0.9522 | 0.9517 | 0.9331 | 0.9342
> | 7B PAPO-G w/ Random-Masking  | 0.9457 | 0.9472 | 0.9525 | 0.9530
> | 7B PAPO-G w/ Semantic-Masking  | 0.9401 | 0.9403 | 0.9513 | 0.9512
>
> As shown in the table above, we find that the overall attention patterns **do not differ significantly** between the two masking strategies. However, the random-masking model **generally exhibits slightly higher entropy**, indicating a more scattered attention distribution. This observation aligns with our hypothesis that semantic masking encourages the model to focus more heavily on a centralized salient region.
>
> ## W1.2: Findings from error analysis:
> We also investigate **what types of errors cause the semantic-masking model to underperform relative to the random-masking model**. We sample 30 instances per dataset from Geo3K, Counting, and MathVerse where the random-masking model is correct but the semantic-masking model fails, and manually categorize the error types.
>
> | Semantic-masking Failure Modes | Perception Error  | Reasoning Error  | Inconsistency Error  | Calculation Error |
> |-|-|-|-|-|
> Geo3K | 73.33% | 6.67% | 3.33% | 16.67%
> Counting | 93.33% | 0.00% | 6.67% | 0.00%
> MathVerse | 76.67% | 6.67% | 6.67% | 10.00%
>
> We find that the **predominant error source is still perception-related errors**, suggesting that the primary advantage of random-masking over semantic-masking lies in improved perception ability.
>
> We believe this is still an open question and a promising topic for future exploration.
>
> ### Reference for W1:
> - [1] He, Kaiming, et al. "Masked autoencoders are scalable vision learners." Proceedings of the IEEE/CVF conference on computer vision and pattern recognition. 2022.
> - [2] Xie, Zhenda, et al. "Simmim: A simple framework for masked image modeling." Proceedings of the IEEE/CVF conference on computer vision and pattern recognition. 2022.
> - [3] Fu, Tsu-Jui, et al. "An empirical study of end-to-end video-language transformers with masked visual modeling." Proceedings of the IEEE/CVF Conference on Computer Vision and Pattern Recognition. 2023.

---

> > ### Author Response · Authors · 2025-11-21
> >
> > # W2.1: Further analysis of PAPO’s effectiveness on low vision-dependency tasks
> > We fully acknowledge the reviewer’s concern. As also discussed in Section 7, the current method applies the implicit perception loss to all instances and all tokens. Although this minimalist design works empirically well, we agree that there is substantial room for improvement through a more sophisticated gating or token-selection mechanism to determine which tokens or instances should receive the loss.
> >
> > To further examine whether the current PAPO model remains effective in strictly vision-independent scenarios, we **conducted an additional evaluation experiment**. Specifically, we **take a widely used text-only benchmark, MMLU-pro [1], and insert dummy visual inputs into the context** (images containing pure random noise). This setup **ensures that the visual input is entirely uninformative.** If the model were to attend to the noisy visual tokens indiscriminately, we would expect a degradation in performance.
> >
> > The results comparing PAPO and GRPO baselines are shown below:
> >
> > | Model (Qwen2.5-VL) | MMLU-Pro + Dummy Visual: Acc (%) |
> > |-|-|
> > | 3B Base | 23.23 |
> > | 3B GRPO | 39.34 |
> > | 3B PAPO-G | 40.74 |
> > |
> > | 7B Base | 31.57 |
> > | 7B GRPO | 50.76 |
> > | 7B PAPO-G | 50.55 |
> >
> > We find that **PAPO still achieves competitive or stronger performance under this extreme setting**, indicating good robustness in low vision-dependency scenarios.
> >
> > ### Reference for W2.1:
> > [1] Wang, Yubo, et al. "Mmlu-pro: A more robust and challenging multi-task language understanding benchmark." Advances in Neural Information Processing Systems 37 (2024): 95266-95290.
> >
> >
> > # W2.2: Further analysis on Figure 1: Does reducing perception errors lead to other error types?
> > As suggested, we conducted an additional qualitative analysis by sampling 60 instances that originally resulted in perception errors under the GRPO baseline and examining their outcomes under PAPO.
> >
> > | Perception Error ->  | Correct | Perception Error  | Reasoning Error  | Inconsistency Error  | Calculation Error |
> > |-|-|-|-|-|-|
> > GRPO | 0 | 100% | 0 | 0 | 0
> > PAPO-G | 41% | 24% | 17% | 8% | 10%
> >
> > As shown in the table, 41% of the original perception errors are corrected, 24% remain perception errors, and 25% shift to other error categories. This indicates that eliminating perception errors alone may not be sufficient for some challenging instances. However, overall, PAPO achieves a substantial 40% complete error removal, demonstrating strong effectiveness in improving the overall performance.

---

### Official Review · Reviewer_Tryh · 2025-11-03

**Soundness:** 3
**Presentation:** 2
**Contribution:** 2
**Rating:** 6
**Confidence:** 3

**Summary:**

- This paper introduces PAPO for multimodal reasoning to encourage visually grounded generation by modifying the core optimization objective, addressing the finding that perception errors on models trained with standard RLVR account for 67% of failures.
- PAPO incorporates an Implicit Perception Loss, a KL divergence term that maximizes the difference between output distributions conditioned on original versus masked visual inputs, combined with a Double Entropy Loss for regularization.
- The algorithm, which requires no additional data or reward models, is integrated into GRPO and DAPO frameworks and evaluated on several benchmarks, demonstrating consistent performance improvements and reduction in perception errors.

**Strengths:**

- This work addresses an important problem of perception error in multimodal reasoning, starting from a clear, evidence-based motivation to achieve practical performance gains.
- The method effectively combines known techniques, such as visual input masking and entropy regularization, with existing RLVR frameworks (GRPO/DAPO) in a simple yet potent way, yielding significant benefits without making the framework overly complex.
- The proposed methodology is validated by extensive analyses, covering hyperparameter impacts, masking strategies, and training dynamics, including a transparent investigation of the failure mode.

**Weaknesses:**

- My main concern is the incomplete experimental comparison to establish its state-of-the-art significance. The paper references a number of existing works that modify GRPO/DAPO for multimodal reasoning, yet the experiments in Table 1 only compare PAPO against the original GRPO and DAPO baselines. Without a comparative analysis against other contemporary SOTA variants under identical training conditions, experimental significance seems limited.
- The framework's generalizability is undermined by its exclusive reliance on the Qwen2.5 model family. While this limitation is acknowledged in Section 7, it is difficult to determine if this is a broadly effective approach or an architectural artifact of Qwen.
- The paper's readability and logical flow are hampered by its over-reliance on the appendix. Critical analyses that form the core of the method's justification are deferred from the main text. This forces the reader to constantly switch contexts to verify the paper's central claims on stability and robustness. For the self-containedness of the main paper, consider aggregating key experimental highlights from the Appendix as a concise figure.

**Questions:**

Please refer to the weaknesses section for detail. Relatively minor questions:
- Considering the benchmarks are largely skewed toward math and geometry, it would be interesting to see how an OCR-based masking strategy (low masking ratio but information-wise critical) would perform with PAPO.
- While the Double Entropy Loss plays a crucial role in stabilizing PAPO, its own sensitivity to hyperparameter choice is not explored.

---

> ### Author Response · Authors · 2025-11-21
>
> **We thank reviewer Tryh for the detailed and insightful comments. We appreciate the reviewer’s acknowledgement of our strengths and for contributing to a positive and professional discussion. We address the remaining questions and suggestions below. We will update the manuscript reflecting these changes before Dec 2.**
>
> # W1: Comparison with other multimodal RLVR modifications
> We agree with the reviewer that including additional comparisons and discussion with other contemporary methods would further strengthen the paper.
>
> We would like to first reiterate that PAPO is motivated by a novel optimization perspective for improving GRPO/DAPO, distinct from prior work that focuses on the reward, rollout, or data perspectives. To clearly isolate and highlight the unique contribution of PAPO, we keep our main comparison centered on the original RLVR base algorithms without introducing other modifications.
>
> However, we find it necessary and interesting to further explore whether PAPO is complementary to the advancements from other perspectives. As suggested, we include a **new experiment considering NoisyRollout [1]**, a contemporary work aimed at improving multimodal RLVR by modifying the rollout stage. The base model is Qwen-2.5-vl-3B.
>
> | Model (Qwen2.5-VL-3B) | Geo3K | MathVista | We-Math | MMK12 | MathVerse | LogicVista | Counting | MMMU-Pro | MathVerse-V | AVG
> |-|-|-|-|-|-|-|-|-|-|-|
> | GRPO | 28.72 | 59.34 | 58.90 | 57.24 | 55.25 | 38.14 | 55.81 | 25.66 | 52.26 | 47.92
> | PAPO-G | 30.95↑ | 61.38↑ |60.09↑ |57.39↑ | 57.14↑ | 38.67↑ | 62.56↑ | 27.11↑ | 53.95↑ | 49.92↑
> | GRPO + NoisyRollout | 28.49↓ | 61.34↑ | 58.58↓ | 66.78↑ | 61.34↑ | 37.47↓ | 61.44↑ | 28.79↑ | 51.30↓ | 50.61↑
> | PAPO-G + NoisyRollout | 30.66↑ | 58.83↓ | 60.60↑ | 66.95↑ | 56.21↑ | 39.51↑ | 71.38↑ | 29.47↑ | 53.36↑ | 51.89↑
>
> We summarize our findings below:
> - The improvements brought by NoisyRollout across datasets are **less consistent** (large gains on some datasets but limited or no improvement on others). Overall, NoisyRollout applied to the 3B GRPO baseline achieves improvements on 5 out of 9 benchmarks, whereas PAPO-G (3B) yields consistent improvements on all benchmarks.
> - PAPO provides **additional gains when combined with NoisyRollout**, demonstrating strong compatibility and highlighting the promise of PAPO as an approach that can be integrated with diverse recent advancements from the reward, rollout, and data perspectives.
>
> ### Reference for W1:
> - [1] Liu, Xiangyan, et al. "Noisyrollout: Reinforcing visual reasoning with data augmentation." arXiv preprint arXiv:2504.13055 (2025).

---

> > ### Author Response · Authors · 2025-11-21
> >
> > # W2: Extend to models other than the Qwen2.5-VL
> > We understand the reviewer’s concern, which also echoed our discussion in the limitations section. We chose to focus on the Qwen2.5-VL family because it is currently the most widely used and infrastructure-supported model series, and it has been adopted as the only base model in the majority of recent multimodal RLVR literature [1,2,3,4,5]. Therefore, we believe that conclusions drawn from experiments on Qwen2.5-VL remain robust and well-supported by the community.
> >
> > Nevertheless, we agree that including results on broader model series would further enhance the evidence of the effectiveness. Thus we try to apply PAPO to Qwen3-VL and InternVL3 models:
> >
> > **New results on Qwen3-VL**: We conduct experiments on the newly released Qwen3-VL-2B-thinking model. Architecturally, Qwen3-VL differs from Qwen2.5-VL through the use of the DeepStack strategy in its vision encoder and a new Interleaved-MRoPE positional embedding.
> >
> >
> > | Model (Qwen3-VL-2B-thinking) | Geo3K | MathVista | We-Math | MMK12 | MathVerse | LogicVista | Counting | MMMU-Pro | MathVerse-V | AVG
> > |-|-|-|-|-|-|-|-|-|-|-|
> > | GRPO | 39.29 | 53.58 | 57.12 | 47.71 | 47.98 | 29.84 | 80.13 | 20.51 | 45.41 | 46.84
> > | PAPO-G | 41.08↑ | 56.08↑ | 59.17↑ | 48.57↑ | 51.89↑ | 32.83↑ | 80.63↑ | 23.42↑ | 50.05↑ | 49.30↑
> >
> > As shown in the Table, we see consistent improvements against the GRPO baseline, demonstrating a strong generalization capability to different model architectures.
> >
> > **InternVL3 attempt**: We also attempted to incorporate the InternVL3 series into our codebase, which is not yet supported by the EasyR1 framework. However, we encountered kernel-related errors that we are unlikely to resolve within the rebuttal period. We will continue reaching out to the EasyR1 and vLLM maintainers to further investigate the issue.
> >
> > ### Reference for W2:
> > - [1] Liu, Xiangyan, et al. "Noisyrollout: Reinforcing visual reasoning with data augmentation." arXiv preprint arXiv:2504.13055 (2025).
> > - [2] Wang, Haozhe, et al. "Vl-rethinker: Incentivizing self-reflection of vision-language models with reinforcement learning." arXiv preprint arXiv:2504.08837 (2025).
> > - [3] Yao, Huanjin, et al. "R1-ShareVL: Incentivizing Reasoning Capability of Multimodal Large Language Models via Share-GRPO." arXiv preprint arXiv:2505.16673 (2025).
> > - [4] Yang, Yi, et al. "R1-onevision: Advancing generalized multimodal reasoning through cross-modal formalization." arXiv preprint arXiv:2503.10615 (2025).
> > - [5] Xia, Jiaer, et al. "Visionary-r1: Mitigating shortcuts in visual reasoning with reinforcement learning." arXiv preprint arXiv:2505.14677 (2025).
> >
> >
> > # W3: Writing improvement: over-reliance on the appendix
> > We thank the reviewer for the suggestion. We will aggregate more key information from the appendix and make use of the additional page in the main text. These changes will be reflected in the updated manuscript.

---

> ### Author Response · Authors · 2025-11-21
>
> # Q1: Additional benchmarking on OCR task & Discussion related to OCR-based masking strategy
>
> The reviewer points out a skewness in our current evaluation benchmarks. To further diversify the evaluation, we **additionally include an OCR-related task: OCR-Bench-v2 [1]**. We report the scores comparing the Qwen2.5-VL-3B and 7B base models, the GRPO baselines, and PAPO-G as follows:
>
> | Model (Qwen2.5-VL) | OCR-bench-v2: Acc (%) |
> |-|-|
> | 3B Base | 20.14 |
> | 3B GRPO | 26.32 |
> | 3B PAPO-G | 27.22 |
> |
> | 7B Base | 27.75 |
> | 7B GRPO | 31.17 |
> | 7B PAPO-G | 32.11 |
>
> We verified that PAPO also brings benefits on such information-dense tasks which require fine-grained perception on the visual inputs.
>
> The reviewer also raises an interesting idea: using OCR as a tool to determine which regions to mask during the masking phase. The motivation is that textual information in an image, especially for math and geometry problems, is often more informative and important. We appreciate this suggestion and believe there is additional nuance to consider. For example, in cases where OCR detects no text, how should masking be handled? Should we fall back to random masking, and how should we balance the variance in masking ratios? We agree that this represents an interesting alternative to random masking and semantic-aware masking. Due to limited rebuttal time, we leave a thorough exploration of this direction to future work.
>
> ### Reference for Q1:
> [1] Fu, Ling, et al. "Ocrbench v2: An improved benchmark for evaluating large multimodal models on visual text localization and reasoning." arXiv preprint arXiv:2501.00321 (2024)
>
>
> # Q2: Present the details on Double Entropy Loss hyperparameter choice ablation
> As suggested, we provide **additional results** on Qwen2.5-VL-7B with PAPO (implicit perception loss coefficient = 0.02) while **varying the double-entropy loss coefficient in 0.01, 0.03, 0.05.** Here, a smaller coefficient means less regularization strength. At 0.01, we observe model collapse in the later stages of training, whereas at 0.03 and 0.05 the entropy loss successfully prevents hacking behavior. Empirically, 0.05 performs the best across 9 evaluation benchmarks.
>
> | Model (Qwen2.5-VL-7B) | Geo3K | MathVista | We-Math | MMK12 | MathVerse | LogicVista | Counting | MMMU-Pro | MathVerse-V | AVG
> |-|-|-|-|-|-|-|-|-|-|-|
> | GRPO | 40.18 | 65.48 | 68.12 | 72.26 | 66.51 | 45.62 | 73.94 | 35.17 | 61.71 | 58.78
> |
> | PAPO-G + Ent Loss 0.01 (collapsed) | 39.95 | 62.03 | 65.19 | 69.71 | 62.72 | 44.30 | 92.50 | 38.01 | 59.77 | 59.35
> | PAPO-G + Ent Loss 0.03 | 40.00 | 68.09 | 67.59 | 72.43 | 68.28 | 46.46 | 89.94 | 35.85 | 61.17 | 61.09
> | PAPO-G + Ent Loss 0.05 | 40.25 | 69.53 | 66.79 | 72.52 | 68.43 | 46.07 | 89.81 | 36.63 | 64.97 | 61.66

---

### Author Response · Authors · 2025-11-24
**Summary of updates in the revised manuscript**

**We thank all the reviewers and the AC for their valuable contributions during the discussion period. Below, we summarize the updates in the revised manuscript, which we have just submitted.**

### Additional Results and Analysis:
- Results with NoisyRollout and its integration with PAPO: **Section 5.1, Table 2**
- Results on Qwen3-vl model: **Table 1**
- Results on OCR-bench-v2: **Appendix K, Table 12**
- Evaluation results on a strictly vision-independent task (MMLU-pro + dummy visual input): **Appendix L, Table 13**
- Ablation on Double Entropy loss coefficients: **Appendix I, Table 11**
- Analysis on attention pattern: **Appendix E, Figure 7**
- Further analysis on masking strategy: **Appendix G.3, Table 9, Figure 8**
- Further analysis on perception error reduction: **Appendix N, Figure 15**

### Writing improvements:
- Moved Figure 5 from Appendix to the main text. Clearly labeled if a Figure/Table is in the Appendix. Ensured that the key takeaways are presented in the main text without requiring readers to constantly jump to the Appendix.
- Added a comprehensive comparison table of related work: **Table 15**

---

### Author Response · Authors · 2025-12-03
**Summary of Rebuttal Discussion**

Dear Area Chair,

Thank you for your time and effort in reviewing our submission. Following the recommendation of the ICLR 2026 Program Chairs, we would like to provide a brief factual summary of the author-reviewer discussion. **All interactions happen prior to the incident.**
**Our goal is to accurately reflect the reviewers’ active engagement during the process, which we greatly appreciate.**

- **All reviewers** provided detailed and professional comments, which we carefully considered. We conducted additional experiments and analyses as described in our previous overall comment.
- **Reviewer zdmV** responded to our initial rebuttal on Nov 21 with additional questions. We subsequently added further responses and visualizations in the revised submission. On Nov 26, reviewer zdmV acknowledged that the revision addressed their questions and raised the score from 8 to 10.
- **Reviewer BkD7** responded to our initial rebuttal on Nov 24 and confirmed that all of their concerns had been fully addressed.


We sincerely thank all reviewers for their tremendous efforts throughout the discussion period, which truly helped us strengthen the submission.

---

### Meta-Review · Area_Chair_kfR2 · 2025-12-27

**Summary:**

This paper proposes PAPO, a novel policy gradient algorithm designed to address the critical bottleneck of visual perception errors in multimodal reasoning with Reinforcement Learning with Verifiable Rewards (RLVR). PAPO introduces Implicit Perception Loss to encourage visually grounded reasoning and Double Entropy Loss for training stability, requiring no additional annotations, reward models, or teacher models. It seamlessly integrates with mainstream RLVR algorithms (e.g., GRPO, DAPO) and delivers significant improvements: 4.4%-17.5% across diverse multimodal benchmarks, 8.0%-19.1% on vision-heavy tasks, and a 30.5% reduction in perception errors.

Reviewers’ core concerns are mainly about the empirical evaluation, such as extending the result to different backbone models, comparing with more baselines, and conducting further analysis and ablation studies.

Overall, PAPO addresses a well-identified bottleneck in multimodal RLVR with a simple yet effective design, delivering substantial empirical improvements and avoiding reliance on additional resources. While it is not theoretically groundbreaking, its practical impact, seamless integration with existing frameworks, and rigorous benchmarking make it a valuable contribution. The outstanding concerns about generalizability and baseline comparisons are minor relative to the core contribution. Considering the overall positive post-rebuttal scores, I recommend acceptance.

**Reviewer Concerns:**

Most of the concerns of reviewers are directly addressed by the authors.

**Reviewer Scores:**

Reviewer zdmV and Reviewer BkD7 confirmed that their concerns are addressed, and would possibly increase the rating.

Reviewer Tryh and Reviewer XvCB didn't respond directly, but most of the concerns are addressed by the authors in my opinion.

---

### Decision · Program_Chairs · 2026-01-26

Accept (Poster)